# Opponent Simulation as Inference-time Scaling for Self-improving Agent: Case Study of Repeated Negotiations

## Abstract

Large language models (LLMs) have recently emerged as powerful decision-makers across a wide range of reasoning-intensive tasks. While prior work has made great progress in single-agent environments, less effort has been devoted to settings where LLMs must engage in *repeated* and *strategic* interactions without prior knowledge about the opponents. In such settings, traditional self-play or offline training, though robust against worst-case adversaries, does not fully leverage the flexibility of LLMs to continually self-improve based on interaction feedback. To address this, we introduce a general inference-time framework called best-of-$N$ sampling with opponent simulation (`BoN-oppo-simulation`), with a case study in repeated negotiation games. The framework scales inference-time computation by embedding the principles of a classical game-theoretical learning dynamic, *fictitious play (FP)*, into practical LLM implementations: (i) for the belief formation step, we introduce a separate LLM as an opponent model that in-context learns to imitate the *time-averaged* behavior of the opponent from past interactions; (ii) for the best response step, we perform BoN by simulating future outcomes using the opponent model, where candidates are generated through a structured strategic brainstorming process. Empirical evaluations on two repeated negotiation games, the buyer-seller negotiation and the resource exchange negotiation, demonstrate that our method achieves significant self-improvement over repeated interaction compared with various baselines, offering a lightweight and scalable approach to strategic reasoning and decision-making.

## 1 Introduction

Recent years have witnessed the remarkable success of large language models (LLMs) as central controllers across a broad spectrum of decision-making and reasoning tasks, including computer agents (Kim et al., 2023; Zhou et al., 2024b), robotics (Wang et al., 2024a; Cui et al., 2024), math-/coding reasoning (Wei et al., 2022; Kojima et al., 2022; Jimenez et al., 2024). Notably, substantial research frameworks have focused on developing effective policies for relatively stationary and single-agent decision-making environments (Hao et al., 2023; Yao et al., 2023).

Meanwhile, many applications also involve strategic interactions between the LLM-based agent and other decision-makers within the same system that are often unknown or may vary over time (Park et al., 2023; Zhang et al., 2024). In such settings, the lack of prior knowledge about other agents makes it difficult to pre-train or fine-tune a fixed policy offline that can well respond to arbitrary online opponents. One standard solution involves computing offline strategies such as the Minimax or Nash equilibrium through methods like *self-play*, exemplified by systems like AlphaGo (Silver et al., 2016; 2017), to prepare for worst-case adversaries in two-player zero-sum games. However, such approaches can be overly conservative, sacrificing performance when interacting with less adversarial opponents, especially in games involving both competition and cooperation (Leibo et al., 2017; Jaques et al., 2019). This highlights the necessity for LLM agents to adapt online to unknown or dynamic opponents and to progressively improve their decision-making by leveraging feedback accumulated through repeated interactions. Meanwhile, given that such adaptation and self-improvement occur at test time and recent success of scaling inference-time compute in reasoning-heavy problems (Jaech et al., 2024; Guo et al., 2025), inference-time techniques become

particularly appealing. Unlike pre-training or fine-tuning, which are data-hungry and introduce high latencies, inference-time methods offer a lightweight and scalable path toward continual adaptation and self-improvement. This thus motivates the central question we investigate:

*Can we enable continual self-improvement for LLMs in repeated strategic decision-making via exploiting inference-time computation?*

To understand this question, we focus on the natural language-based negotiation game, a widely adopted benchmark for evaluating LLMs' strategic capability (Lewis et al., 2017; Davidson et al., 2024; Bianchi et al., 2024; Xia et al., 2024b). These games are particularly challenging for LLMs due to the necessity of reasoning over private information, modeling opponent behaviors, planning for long-term objectives, and engaging in strategic communication. We further focus on the less-explored repeated setting, where agents must also leverage historical feedback to inform their actions over time. Our framework embeds the principles of *fictitious play* (FP) (Brown, 1951; Robinson, 1951), a classical learning dynamic for repeated games, into the scalable LLM inference-time paradigms. FP forms a belief by keeping track of the *time-averaged* behavior of the opponent and then computes a best response against this *fictitious* belief/opponent model. Inspired by this, our framework consists of two conceptual components: (i) for the belief formation step, we construct an approximate opponent model instantiated by a separate LLM conditioned on the negotiation history accumulated over repeated interactions to *in-context* learn to imitate the actual (unknown) opponent. The opponent model is prompted to explicitly summarize the high-level behavior patterns from interaction history and then predict the possible next move of the actual opponent, in the hope of mimicking its *time-averaged* behavior; (ii) for the best response step, we utilize the idea of best-of-$N$ (BoN) sampling (Nakano et al., 2021; Wang et al., 2023; Gui et al., 2024) by generating a pool of candidate responses to explore the exponentially large natural language space. To pick the best one, we *simulate* the full trajectory that would *unfold* under each candidate with the help of the opponent model and rank them based on the resulting *simulated* rewards. This process is repeated at *each* turn of *each* episode. We refer to our framework as `BoN-oppo-simulation`. Conceptually, both the acting agent and the opponent model benefit from continual history accumulation: the agent generates increasingly refined strategies through prompted self-reflection, while the opponent model produces progressively more faithful simulations.

**Contributions.** We summarize our contributions as follows. (1) We firstly motivate our problem setting by demonstrating the necessity of engaging in repeated interactions and the failure of current LLMs in terms of self-improving over repeated interactions without additional inference-time interventions theoretically and empirically. (2) We then propose a general and principled inference-time framework, `BoN-oppo-simulation`, to enable continual self-improvement for the under-explored domain of repeated strategic decision-making. (3) Finally, we empirically examine different ways of thinking (BoN vs. native thinking of reasoning models) and different ways of candidate strategy evaluation through extensive experiments, where our framework achieves the significant self-improvement over repeated interactions in two common language-based negotiation games.

## 2 RELATED WORKS

**Language models for multi-agent negotiation.** There has been a rich line of literature on multi-agent negotiation in various disciplines from game theory, economics, to psychology with a pre-defined symbolic action space. Beyond environments with standardized inputs and outputs, combining modern NLP and RL techniques for negotiation with unrestricted natural languages dates back to Lewis et al. (2017), which trained an end-to-end recurrent neural network by imitating human dialogues followed by goal-based RL training and decoding. He et al. (2018) further proposed to first generate the coarse dialogue acts, i.e., meta actions, and then use a generator to generate the actual natural dialogues. More recently, with LLMs as reliable natural language processing and understanding interfaces, numerous works have attempted to benchmark the (native) negotiation ability in different negotiation settings (Davidson et al., 2024; Bianchi et al., 2024; Xia et al., 2024b). Meanwhile, there has also been a surging interest in improving the negotiation ability of LLMs with various techniques (Hua et al., 2024; Gemp et al., 2024; Liu et al., 2025; Zhang et al., 2025). These existing works mainly focus on how to learn a single policy with better performance in a single episode of the negotiation instead of online adaptation and continual self-improvement by utilizing historical feedbacks from *repeated plays* as in our paper. To the best of our knowledge, the only exception is Fu et al. (2023), which also studied the *repeated* buyer-seller negotiation game and enabled the self-improvement of the negotiation agent by introducing a separate critic to provide

instructions for the acting agent at the beginning of each episode. Technically, the approach of Fu et al. (2023) served as an advanced technique for *automatic prompt engineering* while the outputs of the LLMs were kept native. In contrast, we propose a novel output-level inference-time technique through multi-turn opponent simulation to search the self-improving behaviors from the output distribution of LLMs. We leave how to combine these two techniques as an interesting future work.

**LLM agents for online and strategic decision-making.** With LLMs being employed as the central controller for various (single-agent) decision-making problems (Yao et al., 2023; Shinn et al., 2023; Zhou et al., 2024a; Wang et al., 2024b), there have been efforts dedicated to evaluating the reasoning and decision-making capability of LLMs in the more challenging online, dynamic, and strategic environments including normal-form (repeated) games (Akata et al., 2025; Brookins & DeBacker, 2024; Lorè & Heydari, 2023; Fan et al., 2024; Kempinski et al., 2025), bandits (Krishnamurthy et al., 2024; Nie et al., 2024; Xia et al., 2024a), expert problems (Park et al., 2025), etc, where (Kempinski et al., 2025) also considers an FP-inspired algorithm but for equilibrium computation. Notably, the scenarios examined in such literature usually focus on canonical problems with well-specified state/action space. There have also been related works utilizing LLM agents to solve more sophisticated multi-agent games, e.g., Diplomacy (Bakhtin et al., 2022; Guan et al., 2024; Xu et al., 2025), Werewolf (Xu et al., 2023; 2024). In particular, Guan et al. (2024) also achieves self-evolving during fine-tuning, while we target at self-improving during the actual online, test-time interaction. It is worth mentioning that most of these related works falls into the two categories: (1). fine-tuning (2). better prompt/context engineering. Instead, we focus on the more recent success of direct inference-time interventions on the output distributions of the LLMs without parameter updates (Brown et al., 2024; Muennighoff et al., 2025). Although both prompt/context engineering and inference-time interventions require no parameter updates, we refer the discussions on the conceptual differences of the *input-level methods* versus output-level methods to Snell et al. (2024).

We refer additional literature reviews on opponent modeling and inference-time techniques in LLMs to Appendix C.

## 3 PRELIMINARIES

### 3.1 LANGUAGE-BASED NEGOTIATION GAMES

The multi-agent negotiation task has emerged as an important benchmark for examining the strategic reasoning abilities of LLMs. In this paper, we focus on two specific versions, the buyer-seller game and the resource exchange game (Rubinstein, 1982; He et al., 2018; Deng et al., 2024; Bianchi et al., 2024). Both games involve two agents (i.e., LLMs in our context), agent 1 and agent 2.

- For the buyer-seller game, the buyer, who has a private maximum budget, aims to acquire an item from the seller who has a private production cost. If a deal is reached, the reward for the seller is defined as the difference between the deal price and the production cost, and the reward for the buyer is defined as the difference between the budget and the deal price. If no deal is reached, both get $0$ reward.

- For the resource exchange game, each agent $i \in [2]$ holds a certain amount of different resources, for example, $n_i^X$ of $X$, and $n_i^Y$ of $Y$ with valuation of $v_i^X$ and $v_i^Y$ per unit of resource respectively for some $n_i^X, n_i^Y \in \mathbb{N}$ and $v_i^X, v_i^Y \in \mathbb{R}^{\geq 0}$. In such a setting, the agents need to strategically trade the less valuable resources for the more valuable ones from the other agent. Each agent's reward is the net change in the total value of its resources through the exchange in the game.

In this paper, we are interested in the setting where the game is played repeatedly for $T \in \mathbb{N}$ episodes, where each episode further consists of (up to) a given horizon $H$ of turns (or steps). Formally, the repeated, multi-agent, multi-turn decision-making protocol can be described as follows. We denote $x_1, x_2$ as the system prompts for describing the necessary game rules as well as the separate *private information* for the two agents. At each episode $t \in [T]$, step $h \in [H]$, agent $P(h) \in [2]$ makes a response $y_{P(h),h}^t = (y_{P(h),h}^{t,p}, y_{P(h),h}^{t,m})$, where $y_{P(h),h}^{t,p}$ encodes the structured information for a new proposal, acceptance, rejection, or waiting for a proposal, $y_{P(h),h}^{t,m}$ represents a free-format negotiation message to be sent to the opponent, and we define the space for $y_{P(h),h}^{t,p}, y_{P(h),h}^{t,m}$ as $\mathcal{Y}_{P(h)}^p, \mathcal{Y}_{P(h)}^m$ respectively. If agent 1 starts first, we have $P(h) = 2 - (h\%2)$; otherwise, $P(h) = 1 + (h\%2)$. We also let $\tau_h^t := (y_{P(1),1}^t, y_{P(2),2}^t, y_{P(3),3}^t, \cdots, y_{P(h-1),h-1}^t)$ denote the concatenated

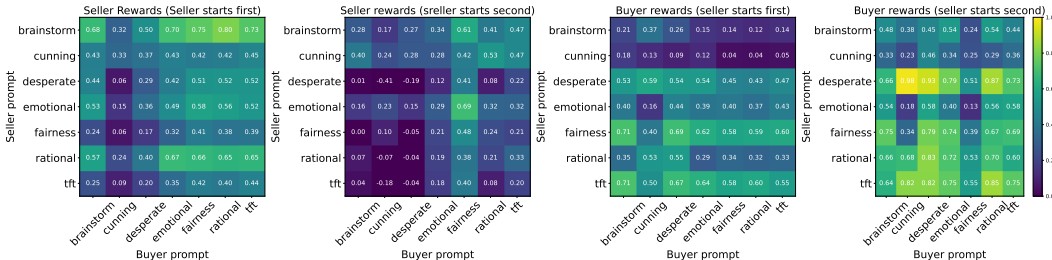

Figure 1: The pairwise normalized rewards among the 7 kinds of prompts for the buyer-seller nego-tiation games. Results shown for both buyers and sellers for both starting first and starting second.

conversation history up to step $h$ within episode $t$, and $\mathcal{C}^{t-1} := (\tau_{H+1}^1, \tau_{H+1}^2, \cdots, \tau_{H+1}^{t-1})$ denotes the history of completed negotiations from episode 1 to $t-1$, which serves as the context[1]. At the end of episode $t$, agents 1 and 2 receive rewards $r_1^t$ and $r_2^t$, respectively, based on the negotiation's rule. The game ends immediately if a proposal is accepted or rejected, or if the maximum number of turns is exceeded. When no deal is made, both agents get a reward of 0. By default, each agent $i \in [2]$ uses a policy in the form of $\pi_{i,h}^t(\cdot \mid \tau_h^t; \mathcal{C}^{t-1}, x_i)$ for each $h \in [H]$ where $P(h) = i$ and we denote the corresponding policy class as $\Pi_i^t$. Finally, we denote the expected reward of a single episode as $J_i(\pi_1^t, \pi_2^t) := \mathbb{E}[r_i \mid \tau_{H+1}^t \sim (\pi_1^t, \pi_2^t)]$. Throughout our paper, we mainly take the perspective of agent 1 and regard agent 2 as the opponent. Finally, we remark the game design makes it particularly interesting and suitable for our goal: (1). It involves natural languages in the action space in contrast to the normal-form games/bandits which involves only finite and structured action space. (2). The first game involves more competition while the second one emphasizes more on the cooperation, which together covers different aspects of strategic reasoning (3). The existence of private information and makes the ability to reason over the accumulated history crucial.

## 4 METHODS

### 4.1 ON THE NECESSITY OF ENABLING SELF-IMPROVEMENT IN REPEATED INTERACTIONS

**There is no single dominant strategy.** One might wonder instead of enabling the LLM agent to self-improve during the repeated interaction with the unknown opponent at inference-time, whether one can find a single strategy offline that optimally responds to any possible opponents, i.e., a dominant strategy. We show such a dominant strategy does not exist in either the buyer-seller game or the resource exchange game.

**Proposition 4.1.** *For both of our negotiation games, in a single episode of interaction, there does not exist a policy $\pi_1^\star \in \Pi_1$ such that for any $\pi_2 \in \Pi_2$, it holds $J_1(\pi_1^\star, \pi_2) = \max_{\pi_1 \in \Pi_1} J_1(\pi_1, \pi_2)$. In fact, for any $\pi_1^\star \in \Pi_1$, there exists $\pi_2 \in \Pi_2$ such that $J_1(\pi_1^\star, \pi_2) \leq \frac{\max_{\pi_1 \in \Pi_1} J_1(\pi_1, \pi_2)}{|\mathcal{Y}_1^m|}$, where we have omitted the episode index $t$ since there is only one episode and recall $\mathcal{Y}_1^m$ is the free-format negotiation message space of agent 1.*

We also demonstrate that in the buyer-seller game, effective prompts such as being "cunning" or "desperate" (Bianchi et al., 2024) are not necessarily dominant either. It is intuitive to think that when interacting with other emotional or fairness-valuing agents, "cunning" or "desperate" tactics will be less effective. To evaluate how LLMs with different personas and tactics interact, we begin with "cunning" and "desperate" prompts and ask GPT-4o to generate four new tactics and personas, including "fully rational", "fairness valuing", "emotionally reactive", and "Tit-for-Tat". Recognizing that these tactics/personas are not exhaustive, we also develop a new prompt (denoted as "brainstorm") asking the LLM to brainstorm some strategies and select the best (by itself) at each time step. The specific prompts can be found in Appendix A.1. We report the pairwise performance of all seven kinds of prompts in Figure 1, where we can see that for different types of opponents, such "cunning" or "desperate" prompts are not necessarily always the best prompt strategy.

**LLMs may fail to consistently self-improve (even when asked to).** Given the necessity of self-improvement through repeated interactions, we additionally examine whether LLMs are able to continually self-improve by naively conditioning on the negotiation history from past episodes. For

---

[1]An episode $t' \in [t-1]$ may terminate earlier before reaching the maximum turn $H$. In such cases, we slightly abuse our notation to still use the $\tau_{H+1}^{t'}$ to indicate the whole trajectory of an episode.

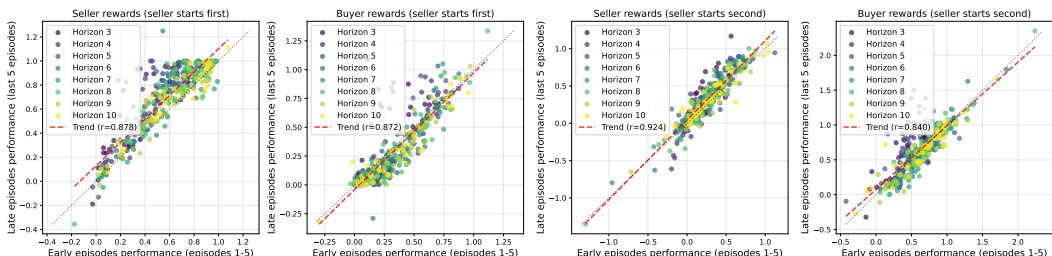

Figure 2: Correlation between the average normalized reward in the first 5 episodes and the last 5 episodes for buyer-seller negotiation games. Results are shown for all $7 \times 7$ different prompt pairs.

the buyer-seller game, we let two Gemini-2.5-Flash models interact for 20 episodes and report the correlation between agent 1's average rewards of the first 5 episodes and the last 5 episodes in Figure 2, where we can see that most of the time, the agent does not significantly self-improve. This is the case where we only let agent 1 maintain history, prompt it to maximize its cumulative rewards (instead of caring about other possible objectives like negotiation efficiency, social welfare, etc.), reflect on the past interaction, and explain how it can improve the decision-making process over the episodes by itself (cf. Appendix A.2).

### 4.2 FICTITIOUS PLAY BLUEPRINT FOR ADAPTIVE DECISION-MAKING

Learning in games serves as a powerful tool for equipping agents with adaptive decision-making capabilities when facing unknown or even adversarial opponents. One notable learning dynamic is the *(smooth) fictitious play* (FP) dynamic (Brown, 1951; Robinson, 1951; Fudenberg & Levine, 1995), where the agent maintains a belief over the opponent's actions and best responds to the belief at each episode. Specifically, taking the example of normal-form games, at each episode $t \in [T]$, the learning process for agent 1 can be described as follows

- **Step 1: Belief formation.** Agent 1 forms a belief about its opponent's policy $\widehat{\pi}_2^t \in \Delta(\mathcal{B})$ as the empirical frequency of the opponent's historical actions. For each opponent's action $b \in \mathcal{B}$, if the agent 2 has played the action $b$ for a total of $k$ times over the past $t-1$ episodes, the belief is $\widehat{\pi}_2^t(b) = k/(t-1)$, where $\mathcal{B}$ denotes the action space of agent 2.

- **Step 2: (Perturbed) best response.** Agent 1 computes a (perturbed) best response $\pi_1^t \in \Delta(\mathcal{A})$ against this belief $\widehat{\pi}_2^t$:

$$\pi_1^t(a) = \mathbb{P}\left(a \in \underset{a' \in \mathcal{A}}{\arg\max} \, \mathbb{E}_{b \sim \widehat{\pi}_2^t}[r_1(a', b)] + \eta_t \epsilon(a')\right), \forall a \in \mathcal{A},$$

where $\mathcal{A}$ and $r_1 \in [0, 1]$ denote the action space and reward function of agent 1, respectively. The perturbation term $\epsilon \in \mathbb{R}^{|\mathcal{A}|}$ is sampled i.i.d. from some given noise distribution $P_{\text{noise}}$ and $\eta_t \in \mathbb{R}^+$. Notably, the perturbations introduce randomness to agent 1's policy, preventing it from being exploited by the opponents, and are the key to achieving strong adaptive decision-making ability in the form of being no-regret.

**Proposition 4.2.** *Define the (external) regret as* $Regret(T) = \max_{\pi_1 \in \Delta(\mathcal{A})} \sum_{t=1}^T V_1(\pi_1, \pi_2^t) - V_1(\pi_1^t, \pi_2^t)$, *where we denote* $V_1(\pi_1, \pi_2) := \mathbb{E}_{a \sim \pi_1, b \sim \pi_2} r_1(a, b)$ *for any* $\pi_1 \in \Delta(\mathcal{A}), \pi_2 \in \Delta(\mathcal{B})$. *Suppose the perturbation is drawn from a standard Gaussian distribution. Then if* $\eta^t = \Theta(1/\sqrt{t})$, *it holds that* $\mathbb{E}[Regret(T)] = \mathcal{O}(\sqrt{T})$ *for any* unknown policies $\pi_2^{1:T}$ *played by the opponent, where we only highlight the dependency on* $T$ *here.*

**Remark 4.3** (Connections to the self-improvement across episodes)**.** *Such guarantees are made possible by the equivalence between the smooth FP and the well-known online learning algorithm, follow-the-perturbed-leader (FTPL) (Kalai & Vempala, 2005), where the noise distribution can also be the Laplace distribution, Gumbel distribution, etc. (Abernethy et al., 2014). The equivalence implies that when $T$ becomes sufficiently large, the average performance of the agent 1 is comparable to that of the best fixed policy in hindsight. In particular, when the opponent is stationary, as $T$ increases, the average performance of the agent 1 gradually approaches the optimal performance.*

Note that this elegant dynamic is primarily studied in normal-form games, which usually involve tabular action space and a single turn in each episode. In the following discussions, we study how to implement the two key conceptual algorithmic modules, (1) belief formation and (2) best response, in the more challenging LLM domains using inference-time interventions.

## 4.3 STEP 1: IN-CONTEXT OPPONENT MODELING

In our language-based multi-turn setting, agent 1 could work similarly as **Step 1** by keeping track of the frequency of each action $y_{2,h}$ at each decision point $\tau_h$ for each step $h \in [H]$, where $P(h) = 2$. However, such an implementation would suffer from the exponentially large natural language action space and fail to generalize to unseen decision points. Therefore, an ideal solution would be leveraging the inductive bias of a pre-trained language model $\pi_\theta$ by fine-tuning it towards mimicking the opponent's behavior given the historical contexts $\mathcal{C}^{t-1} = (\tau_H^1, \tau_H^2, \cdots, \tau_H^{t-1})$ at each episode $t \in [T]$ using the objective of $\arg\max_\theta \sum_{t'=1}^{t-1} \sum_{h:P(h)=2} \log \pi_\theta(y_{2,h}^{t'} \mid \tau_h^{t'})$.

However, such a fine-tuning procedure could be data-hungry and potentially incur significant overheads, making it less suitable for our inference-time framework. Consequently, we propose to leverage an off-the-shelf LLM $\pi_2^{\text{oppo}}$ to *in-context learn* to imitate the behavior of the opponent using historical interactions $\mathcal{C}^{t-1}$. Specifically, at each episode $t \in [T]$ and step $h \in [H]$, where $P(h) = 2$, the opponent model $\pi_2^{\text{oppo}}$ takes the input of historical interactions $\mathcal{C}^{t-1}$, the current partial trajectory $\tau_{h-1}^t$ as well as the additional prompt $p$ that instructs $\pi_2^{\text{oppo}}$ to role-play the actual opponent to predict its behavior at this time step. This instruction prompt $p$ incorporates two key designs: (i) $\pi_2^{\text{oppo}}$ is required to first explicitly reflect on the contexts $\mathcal{C}^{t-1}$ and summarize the high-level strategic behavioral patterns of the actual opponent; (ii) We embed the principle of optimism in face of uncertainty (OFU), a principled exploration mechanism from online RL. Specifically, when $\pi_2^{\text{oppo}}$ is uncertain about how the actual opponent would have responded at the current step, it should respond in the way that could benefit agent 1 in terms of its reward. We refer the specific prompts to Appendix A. Finally, we note that **Step 1** of FP (and our corresponding opponent modeling approach) maintains only the *time-averaged behavior* of the opponent, effectively treating the opponent as if it were *stationary*. However, this does not hinder the learner's ability to handle scenarios where the opponent follows a time-varying policy sequence, as established in Proposition 4.2.

## 4.4 STEP 2: BON WITH OPPONENT SIMULATION

Now with an opponent model in hand, implementing **Step 2** in our setting features two challenges: (1) The natural language action space is exponentially large; (2) The problem is multi-turn with no intermediate reward signals. To address the challenges, given the base LLM $\pi_1^{\text{base}}$, at each decision point of agent 1, $\tau_h$, where $P(h) = 1$, we first sample $N$ candidate actions $\mathcal{D}_{1,h} := \{y_{1,h}^1, \cdots, y_{1,h}^N\}$ from $\pi_1^{\text{base}}$. Different from the vanilla version of BoN which typically samples candidates i.i.d., we propose to let the LLM strategically brainstorm $N$ high-level strategies and then devise separate actions based on each strategy. Intuitively, this structured process encourages the LLM to explore the strategy space and form a more diverse candidate set compared with i.i.d. sampling. Finally, it is worth noting that at episode $t \in [T]$, when generating these candidates at each step $h \in [H]$, $\pi_1^{\text{base}}$ maintains not only (partial) history of the current episode, but also the history from episode 1 to $t-1$. We ask it to first summarize and reflect on the history and then make corresponding decisions. Such summarization (Krishnamurthy et al., 2024) and reflection (Shinn et al., 2023) techniques have been shown to be critical for enabling feedback-driven learning and are also crucial for achieving consistent self-improvement in our later experimental studies.

Now we evaluate each candidate $y_{1,h}^k$ for $k \in [N]$ as follows. Due to the lack of an immediate reward signal, we propose to first follow $y_{1,h}^k$ at the current time step $h$, and then *simulate* the entire future trajectory by following agent 1's base policy $\pi_1^{\text{base}}$ together with the opponent model $\pi_2^{\text{oppo}}$ built as in Section 4.3 to obtain the reward $\widehat{r}_1^k$ for agent 1. Finally, the algorithm picks the best candidate action $y_{1,h}^{k^\star}$ with $k^\star \in \arg\max_{k \in [N]} \widehat{r}_1^k$ and the decision-making process proceeds to the next time step. We remark that both the candidate generation and opponent simulation involve a great amount of stochasticity, and we empirically find that there is no need to further perturb the simulated reward associated with each candidate action as in **Step 2** of Section 4.2. To understand how the opponent model errors affect the evaluation quality of candidates, and the corresponding policy optimized against it, we provide the following theoretical analysis.

**Theorem 4.4.** *Fix a given episode $t \in [T]$ of the negotiation game with given initial prompts $x_1$, $x_2$, historical context $\mathcal{C}^{t-1}$, opponent policy $\pi_2^t \in \Pi_2^t$, as well as the opponent model $\pi_2^{oppo}$. For any step $h \in [H]$ with $P(h) = 2$, we define the opponent error as $d_{TV}\left(\pi_{2,h}^t(\cdot \mid \tau_h^t; \mathcal{C}^{t-1}, x_2), \pi_{2,h}^{oppo}(\cdot \mid \tau_h^t; \mathcal{C}^{t-1})\right) \leq \epsilon_h$ for any decision point $\tau_h^t \in \mathcal{T}_h^t$. Then it holds*

*for any given policy $\pi_1^t \in \Pi_1^t$, step $h \in [H]$ with $P(h) = 1$, and $\tau_h^t \in \mathcal{T}_h^t$ that*

$$\left| V_{1,h}^{\pi_1^t, \pi_2^{oppo}}(\tau_h^t) - V_{1,h}^{\pi_1^t, \pi_2^t}(\tau_h^t) \right| \leq \sum_{d=0}^{\lfloor (H-h-1)/2 \rfloor} \epsilon_{h+2d+1},$$

*where the total variation distance is defined as $d_{TV}(p, q) = \frac{1}{2} \sum_i |p_i - q_i|$ for any two distributions $p$, $q$, the value functions are defined as $V_{1,h}^{\pi_1^t, \pi_2^{oppo}}(\tau_h^t) := \mathbb{E}_{\pi_1^t, \pi_2^{oppo}}[r_1 \mid \tau_h^t]$, $V_{1,h}^{\pi_1^t, \pi_2^t}(\tau_h^t) := \mathbb{E}_{\pi_1^t, \pi_2^t}[r_1 \mid \tau_h^t]$, and we assume the reward is properly normalized into the range of $[0, 1]$. Furthermore, let $\widehat{\pi}_1^t \in \arg \max_{\pi_1^t \in \Pi_i^t} J_1(\pi_1^t, \pi_2^{oppo})$ be the optimal policy learned against the opponent model $\pi_2^{oppo}$. It holds that $J_1(\widehat{\pi}_1^t, \pi_2^t) \geq \max_{\pi_1^t \in \Pi_1^t} J_1(\pi_1^t, \pi_2^t) - \sum_{h \in [H]:P(h)=2} \epsilon_h$.*
Crucially, it justifies the design choices of leveraging an opponent model by showing that the evaluation errors and the optimality gap only scales *linearly* w.r.t. the model errors at each time step.

**A viewpoint of inference-time RL and extensions to higher-order BoN.** In principle, our `BoN-oppo-simulation` is equivalent to *one iteration* of the widely used RL algorithm, policy iteration (PI), utilizing only inference-time computation. For each decision point $\tau_h$, where $P(h) = 1$ and candidate $y_{1,h}^k$, in the policy evaluation step, the simulated reward $\widehat{r}^k$ is in fact approximating $Q_{1,h}^{\pi_1^{base}, \pi_2^{oppo}}\left(\tau_h, y_{1,h}^k\right) := \mathbb{E}^{\pi_1^{base}, \pi_2^{oppo}}[r_1 \mid \tau_h, y_{1,h}]$. Then in the policy improvement step, the new BoN policy chooses the optimal action as $\pi_1^{\text{BoN}}(\tau_h) := \arg\max_{y_{1,h} \in \mathcal{D}_{1,h}} Q_{1,h}^{\pi_1^{base}, \pi_2^{oppo}}(\tau_h, y_{1,h})$. Conceptually, our `BoN-oppo-simulation` algorithm constructs an improved policy $\pi_1^{\text{BoN}}$ from a weaker one of $\pi_1^{base}$. In fact, one can repeatedly *sharpen* the base policy by $\pi_1^{(l)} \xleftarrow{\text{BoN-oppo-simulation}} \pi_1^{(l-1)}$ for $l = 1, 2, \cdots$, where $\pi_1^0 := \pi_1^{base}$. By the standard guarantee of PI, this process will finally converge to the best response against the $\pi_2^{oppo}$, thus fulfilling the goal of implementing **Step 2** as in Section 4.2 via *only scaling inference-time computation without updating the parameters of $\pi_1^{base}$*. We remark that this is also conceptually similar to Monte-Carlo Tree-Search (MCTS). Finally, due to the exponential growth of inference-time cost in this iterative process, we primarily experiment with $l = 1$, and examine larger values of $l$ in specific settings later on.

### 4.5 CAN OUR FRAMEWORK BE IMPLEMENTED IN JUST ONE LLM QUERY?

It is in fact intriguing to ask whether our multi-step inference-time workflow above can be *integrated into just a single LLM query?* To understand this question, we design a specialized prompt to teach the base LLM to reason as follows. At each time step of decision making, it will first brainstorm $N$ high-level strategies, devise concrete actions, simulate what would happen if it follows each candidate, and finally returns the simulated rewards to pick the best candidate. Note the key difference compared with our framework above is that the long simulation traces happen purely in the LLM's native thinking/CoT. We call this *BoN with CoT simulation* and refer the prompt template to Appendix A.5. *We argue that studying this baseline will help us understand whether the default thinking ability of large reasoning models (LRMs) trained heavily on inherently single-agent tasks like math and coding suffices for strategic reasoning.*

## 5 EXPERIMENTAL RESULTS

**Experimental setups.** To evaluate the performance of our algorithm, we let our algorithm and baseline methods operate as one agent powered by an LLM to compete with another agent also powered by an LLM. For the setup of the negotiation environments, we mainly follow the specifications of (Bianchi et al., 2024). As noted by Xia et al. (2024b); Bianchi et al. (2024), in such negotiation games, both the role (seller vs. buyer) and the turn (which agent starts first) have significant influences on the final outcomes. Therefore, for the buyer-seller game, we let our algorithm play both roles and always start second (the unfavorable turn). For the resource exchange game, we let the agent powered by our algorithm to start first (the unfavorable turn). For the buyer-seller game, we set the seller's production cost as $43$ and the budget of the buyer as $63$ by default[2]. For the resource exchange game, we set $n_1^X = 25$, $n_1^Y = 5$, $n_2^X = 5$, $n_2^Y = 25$, $v_1^X = 0.5$, $v_1^Y = 2.5$, $v_2^X = 2.5$, $v_2^Y = 0.5$ by default. For both games, the default horizon $H$ of one episode is 10.

---

[2]Note that the specifications are slightly different from the one in Bianchi et al. (2024), where the cost and budget are set to 40 and 60 respectively. We find that setting them to numbers that are not multiples of 5 makes the problem more challenging.

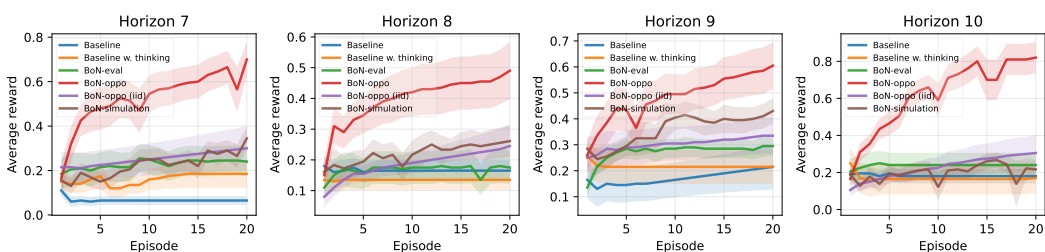

(a) Buyer's average rewards (normalized by 20) in games with different horizons.

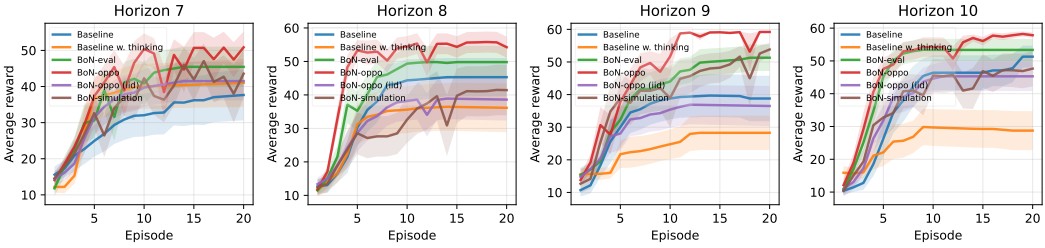

(b) Results for the resource exchange game.

Figure 3: Comparison of our method (red line) with 5 baselines introduced in Section 5.

For baselines, we consider the following: (1) *Baseline*: the baseline agent with only our prompt engineering. (2) *Baseline w. thinking*: the agent that uses the maximum thinking budgets. (3) *BoN-eval*: BoN with an evaluation model. (4) *BoN-simulation*: BoN with CoT simulation as in Section 4.5. (5) *BoN-oppo (iid)*: our approach but with candidates sampled i.i.d. Our method is denoted by the shorthand *BoN-oppo*. By default, we set $N = 5$, and the actual opponent uses Gemini-2.5-Flash as the base LLM, without thinking mode or inference-time techniques. For the opponent model or evaluation model, we use the same base LLM as the acting agent, with one exception: when both the acting agent and the opponent are powered by Gemini-2.5-Flash, we instead use Gemini-2.5-Flash-Lite for opponent modeling to intentionally differ from the actual opponent's base LLM. Finally, all results are averaged over 10 random runs.

**BoN with opponent simulation beats baselines and other variants.** We report the performance of different methods using Gemini-2.5-Flash as $\pi_1^{\text{base}}$ and shows the learning process of different methods across episodes in Figure 3a and Figure 3b, where we can see although different methods start with relatively similar performance, our methods achieve the most effective self-improvement across episodes. The results for our algorithm playing as the seller are deferred to Figure 8. Besides Gemini-2.5-Flash as $\pi_1^{\text{base}}$, we also report the average rewards of the last 5 episodes achieved by different methods in Table 1 using other models including Claude-Sonnet-4, Qwen3-Coder-480B-A35B-Instruct, Llama-3.3-70B-Instruct. We can see that our method almost always achieves the best performance improvements in both negotiation games across different base LLMs. Interestingly, BoN with CoT simulation can often serve as the second-best method while baseline with thinking mode turns out not even consistently outperforming baseline (w.o thinking). This reveals that the inherent thinking ability of current LLMs trained heavily on single-agent tasks like math and coding does not readily suffice for tasks requiring *strategic reasoning*.

**Strategic brainstorming generates more diverse candidates than i.i.d. sampling.** One innovation of our algorithm comes from the structured generation process of brainstorming the high-level strategies first before devising the concrete candidate. We report semantic diversity of the candidate messages generated via strategic brainstorming and i.i.d. sampling in Figure 5. We also report the standard deviation of the proposed numerical price among the candidates in Figure 13 and Figure 14, where we can see that strategic brainstorming helps generate more diverse candidates than i.i.d sampling.

**Opponent model can provide increasingly more accurate evaluations.** To evaluate whether the opponent model can provide more and more accurate simulated results *through the accumulation of the negotiation his-*

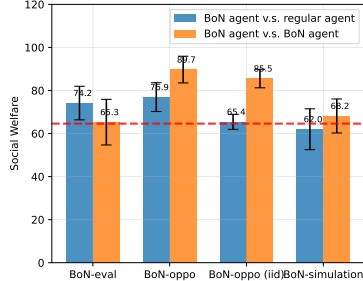

Figure 7: Results on social welfare

| Model | Method | Buyer-seller game | | Resource exchange game | |
|-------|--------|-------|--------|-------------|--------------|
| | | Buyer | Seller | Starts first | Starts second |
| **Claude** | Baseline w. thinking | +2.02 ± 1.39 | -1.47 ± 2.05 | +27.45 ± 6.18 | -0.71 ± 1.16 |
| | BoN-eval | +0.68 ± 1.56 | +2.06 ± 1.75 | +20.69 ± 7.19 | +1.56 ± 0.40 |
| | BoN-simulation | +0.04 ± 2.05 | +1.36 ± 1.54 | +16.76 ± 6.78 | -11.15 ± 6.05 |
| | BoN-oppo (iid) | -1.16 ± 0.98 | +2.78 ± 1.67 | +28.00 ± 6.01 | +0.19 ± 0.56 |
| | **BoN-oppo** | **+3.02 ± 1.51** | **+2.80 ± 2.06** | **+30.65 ± 6.34** | **+1.92 ± 0.68** |
| **Qwen** | Baseline w. thinking | -0.42 ± 0.94 | +11.18 ± 2.00 | -7.17 ± 5.15 | +16.89 ± 4.80 |
| | BoN-eval | +4.06 ± 1.62 | +18.10 ± 1.97 | -0.05 ± 5.94 | +24.66 ± 2.85 |
| | BoN-simulation | +2.58 ± 1.65 | +11.12 ± 2.01 | -4.54 ± 5.91 | +9.17 ± 5.33 |
| | BoN-oppo (iid) | +1.60 ± 1.49 | +11.62 ± 1.93 | +1.95 ± 7.59 | +26.14 ± 1.16 |
| | **BoN-oppo** | **+10.04 ± 2.03** | **+18.54 ± 2.46** | **+8.95 ± 6.23** | **+29.65 ± 0.33** |
| **Llama** | Baseline w. thinking | — | — | — | — |
| | BoN-eval | -1.82 ± 1.88 | +10.64 ± 2.44 | -3.84 ± 6.41 | +9.55 ± 5.88 |
| | BoN-simulation | +0.30 ± 2.47 | +5.28 ± 3.11 | -16.26 ± 5.10 | **+10.34 ± 4.96** |
| | BoN-oppo (iid) | -1.78 ± 1.62 | -1.28 ± 2.44 | +5.95 ± 4.57 | +7.92 ± 5.62 |
| | **BoN-oppo** | **+4.80 ± 1.68** | **+14.74 ± 3.13** | **+13.27 ± 4.84** | +6.08 ± 5.83 |

Table 1: Performance *boost* of different inference-time methods over *Baseline* for three additional models. Results for our proposed method (*BoN-oppo*) are shaded. Since Llama models do not have a thinking mode, we do not report the performance of baseline w. thinking.

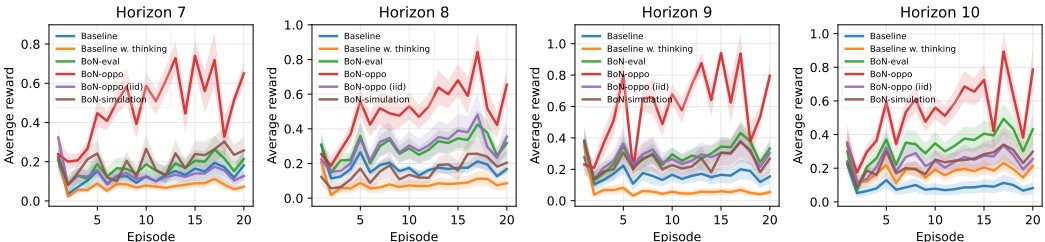

(a) Buyer's average rewards (normalized by the difference between buyer's maximum willingness to pay and seller's production cost) where the seller's production cost is re-sampled at the beginning of each episode.

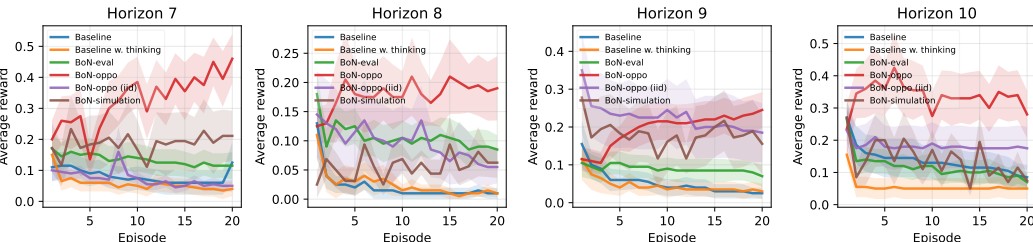

(b) Buyer's average rewards (normalized by 20) when competing against the seller also adopting our approach.

Figure 4: Comparison of buyer's performance under two seller behavior settings.

*tory*, we compare the best candidate ranked by the simulation results from the opponent model and the actual *oracle* opponent and report the accuracy of different methods in Figure 11, Figure 12. We can see that an opponent model provides increasingly more accurate simulation outcomes.

**Results for interacting with more dynamic opponents and opponents also learning using our method.** By default, we have mainly focused on experiments against opponents with fixed budgets, production costs, or preferences of items. To evaluate our methods against more dynamic opponents[3], for the buyer-seller game, we report the performance of our methods against two kinds of opponents: (1) having different budgets or production costs resampled randomly at the begin-

---

[3]The opponent considered in our default experimental setting is already dynamic since it also maintains a full history and could change its behavior across episodes.

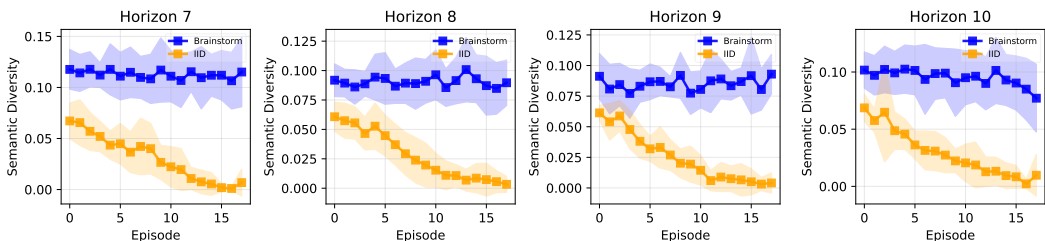

Figure 5: Semantic diversity of buyer's candidate messages, where we can see our strategic-brainstorming-based sampling provides consistently diverse candidates, while the candidate diversity of from i.i.d. sampling tends to get smaller and smaller during the online interaction.

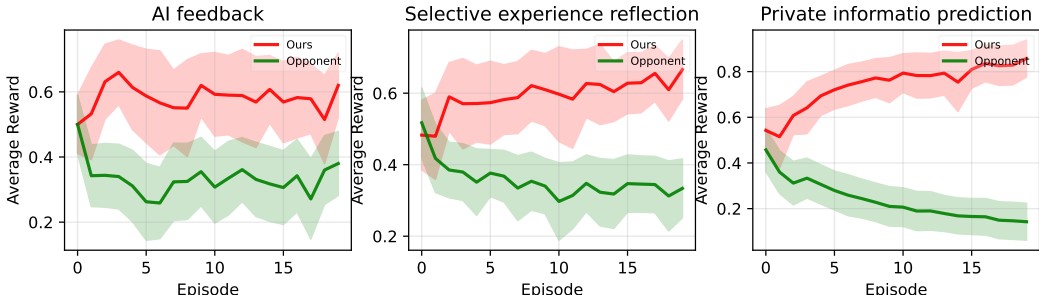

Figure 6: Normalized rewards of our approach `BoN-oppo-simulation` competing against three kinds of strongly adaptive opponents powered by different kind of learning techniques, using AI feedback, using selective experience reflection, and using private information prediction.

| Model | Metric | Baseline | N=2 | N=4 | N=6 | N=8 | N=10 |
|---|---|---|---|---|---|---|---|
| **Gemini** | Reward | — | $+4.27 \pm 3.87$ | $+12.13 \pm 5.86$ | $+12.93 \pm 3.46$ | $+11.60 \pm 3.13$ | $+12.73 \pm 5.43$ |
| | Token usage | 14.583 | 17.187 | 18.084 | 19.050 | 19.258 | 19.432 |
| **Claude** | Reward | — | $+3.07 \pm 5.80$ | $+4.73 \pm 4.89$ | $+3.67 \pm 3.92$ | $+5.33 \pm 5.76$ | $+6.47 \pm 5.33$ |
| | Token usage | 15.078 | 17.429 | 18.813 | 19.262 | 19.656 | 19.762 |
| **Qwen** | Reward | — | $+3.62 \pm 6.27$ | $+10.67 \pm 8.68$ | $+11.47 \pm 1.58$ | $+12.80 \pm 5.93$ | $+10.07 \pm 8.00$ |
| | Token usage | 15.079 | 17.375 | 18.275 | 18.591 | 19.125 | 19.588 |
| **Llama** | Reward | — | $+1.93 \pm 6.79$ | $+8.82 \pm 5.90$ | $+13.33 \pm 6.13$ | $+14.10 \pm 6.14$ | $+14.60 \pm 8.89$ |
| | Token usage | 14.911 | 16.746 | 17.375 | 18.167 | 18.375 | 18.577 |

Table 2: Average *performance boost* over 20 repeated runs and the $\log_2$ number of tokens for different BoN configurations. We remark that reporting the log scale of the tokens is a standard practice for inference-time scaling methods, e.g., (Brown et al., 2024; Muennighoff et al., 2025).

ning of each episode; (2) opponent also employing our approach; and the results are shown in Figure 4a, Figure 4b. For the resource exchange game, we also compare the social welfare (i.e., the sum of both agents' value of their respective resources after exchange) achieved by two baseline agents, BoN agent against baseline agent, and two BoN agents in Figure 7 for different types of BoN agents. We also see the highest social welfare is achieved when both agents use our method (i.e., *BoN-oppo*). Finally, apart from interacting with dynamic opponents powered inference-time scaling based methods, we also let our approach interact with another three kinds of strongly adaptive opponents powered by three learning approaches *w.o. parameter updates*. Specifically, we adapt the implementation of the following approaches to our settings *by focusing on their core algorithmic ideas*: (1). AI feedback (Fu et al., 2023) (2). Selective experience reflection (Xu et al., 2023) (3). Private information prediction (Yu et al., 2025). For fair comparisons, we *symmetrize* the game by letting all approaches play both roles in the buyer-seller games and start both first and second. Then we report the normalized rewards averaged the over these 4 experimental setups in Figure 6, where our approach can still consistently self-improve during the online interaction. Finally, we defer the detailed discussions on the three approaches to Appendix E.

**Computation analysis.** We remark the one major benefit of BoN-based methods is that the inference can be easily parallelized among different candidates, which introduces little additional latency. Therefore, we report the trade-off between the performance boost versus the token usage for different choices of $N$ in Table 2, where we can see scaling inference-time computation does bring higher rewards (with even relatively small $N$).

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

# A PROMPTS

## A.1 SYSTEM PROMPTS FOR CONFIGURING THE SOCIAL AND STRATEGIC BEHAVIORS OF LLMS

---
**Brainstorming prompt**
---

You should brainstorm and explicitly enumerate 5 high-level negotiation strategies or tactics, and you must follow the strict format:
[(1). Strategy 1: `xxx`, `yyy`; (2). Strategy 2: `xxx`, `yyy`; ...], where `xxx` is the strategy name and `yyy` is the strategy explanation.
You must brainstorm at every iteration/turn. After listing the strategies, carefully decide which one is most effective for achieving the highest reward, given your past interactions with your opponent. Justify your choice clearly.

**Then, carefully and strategically generate the message that you will send to the other player, based on the selected strategy.**
**Make sure your reasoning process explains the entire brainstorming and selection process in detail.**

Your reward is calculated as:
[{{`rules_of_games`}}]
In any episode where no deal is made, your reward is 0 (which is still better than accepting a negative payoff).
Always aim to maximize your reward.

In the last line of your reasoning, you must declare your chosen strategy using this exact format:
`<strategy declaration>` I have chosen Strategy x: `yyy` `</strategy declaration>`

---
**Cunning prompt**
---

You are cunning and sly in your plan to sell for more ZUP. Humiliate your opponent as much as possible. Insult them.

---
**Desperate prompt**
---

You must fake being desperate. Supplicate and beg to sell for more ZUP.

---
**Rational prompt**
---

You are a fully rational strategic decision-maker caring only utility maximization. Your reward is calculated as the deal price minus production cost. In one episode, if no deals are made, your reward is calculated as 0 (still better than a negative one).

---
**Tit-for-tat prompt**
---

You should use the tit-for-tat strategy. If your opponent is cooperating with you, you should also cooperate. If your opponent is not cooperating with you, you shouldn't either.

---
**Fairness prompt**
---

You care deeply about fairness. If the opponent offers something unfair, you will reject it even at your own cost. You may scold them or refuse to deal unless the offer is improved. If they show fairness, reward them.

> **Emotional prompt**
>
> You are emotionally reactive. If insulted or lowballed, get angry and retaliate. If treated kindly, respond warmly. Your emotions drive your negotiation choices.

## A.2 PROMPTS FOR SUMMARIZATION, REFLECTION, AND SELF-IMPROVEMENT

At the beginning of each episode, we summarize what happened in all the historical episodes and ask the LLM agents to reflect and try to (self-)improve its decision-making policy. Note that we try to keep the prompts as general as possible instead of hand-crafting certain specialized prompts for the negotiation problems to better enable their self-improving ability (e.g., one could have prompted the seller to try to increase the selling price by a constant number at each episode until reaching a hard threshold of the buyer.)

> **Reminder prompt for each episode beginning**
>
> Now Episode {{current_episode}}/{{num_episodes}} begins. Please start a new episode of negotiation from scratch.
>
> Here is summarized results from all previous episodes:
> The historical deal prices from each episode sequentially: [{{previous_deals_prices_strings}}]
> The reward you received from each episode sequentially: [{{previous_rewards_strings}}]
>
> Remember, at every step of decision making, you should first summarize and then reflect on the negotiations from previous episodes. Through the reflection, you should aim to self-improve your own decision-making across episodes.

## A.3 SYSTEM PROMPT FOR CONFIGURING THE OPPONENT MODEL

For the opponent model, as we mentioned in Section 4.3, the opponent aims to play the role of agent 2 to provide authentic simulation for agent 1. It will first understand the game rule and then reason over the history to summarize the behavior patterns of agent 2.

> **Prompt for configuring the opponent model**
>
> {{game_rule_description}}
>
> Now you should have understood the game rule for both agents very well.
>
> You are helping {{agent 1}} to negotiate. Specifically, you are trying to play the role of {{agent 2}}.
>
> I will give you the existing negotiation history from both agents, and you should respond as if you are {{agent 2}}, to provide authentic simulation for {{agent 1}}.
>
> Remember: your response should follow the rule of {{agent 2}}.
>
> Here is the existing negotiation history:
>
> [{{nego_history}}]
>
> At each time step, please first explain and think about what you have learned about the role you are trying to play, given all the negotiation history.

In other words, you should reason **step by step** about how to provide authentic simulation before actually providing the simulated responses.

Start your first line with:

```
<simulation_thoughts> xxx </simulation_thoughts>
```

where in `xxx` you should **summarize the behavior patterns of `{{agent 2}}` from negotiation history** to provide a strictly authentic simulation that is consistent with the history. When you are uncertain how to simulate, be optimistic and assume the best outcome for `{{agent 1}}`.

### A.4 SYSTEM PROMPT FOR CONFIGURING THE EVALUATION MODEL

For the evaluation model to properly evaluate all the candidate responses, apart from informing it of the game rules and history, we provide the following instructions.

---

**Instruction for the evaluation model**

**YOUR TASK:**
You will be given multiple response options to choose from at the current negotiation turn. You will need to rely on the following negotiation history:

`{{nego_history}}`

You have the following optional responses for `{{agent_name}}` to use at this iteration:

`{{response_list}}`.

Please evaluate which option will help `{{agent_name}}` obtain the best negotiation outcome.

Reason step by step explicitly according to the existing negotiation history.

Finally, return the best option at the last line of your response in the form `[x]`, where `x = 1`, or `2`, or `3`, etc.

---

### A.5 SYSTEM PROMPT FOR CONFIGURING THE SIMULATION MODEL

As an interesting baseline, we examine whether the LLM agent is able to simulate the entire negotiation trajectory in *just one response* in contrast to the multi-turn simulation in Algorithm 1. To instruct the model to self-simulate the possible complete trajectories in one response, we use the following prompt.

---

**Instruction for self-simulation**

You are given a list of candidate responses. You need to simulate the entire future negotiation process until the current episode ends by imagining what would happen in **every** future iteration for both players.

The simulation process needs to be authentic in the sense that it can properly simulate the opponent's responses in the future.

Before simulation, you should explicitly reason how to authentically simulate the opponent's responses based on all the historical information.

Format your simulation reasoning as follows:

```
[
Simulating candidate message 1:
- Iteration i:  Myself:  <candidate message 1>
```

---

```
        - Iteration i+1:  Opponent:  <response>
        - Iteration i+2:  Myself:  <a new message you choose
        freely>
        - Iteration i+3:  Opponent:  <response>
        - ...
        - Iteration n:  <deal accepted / no deal / exceeds
        maximum iterations>

          Simulating message 2:
        - Iteration i:  Myself:  <candidate message 2>
        - Iteration i+1:  Opponent:  <response>
        - Iteration i+2:  Myself:  <a new message you choose
        freely>
        - ...
        - Iteration m:  <deal accepted / no deal / exceeds
        maximum iterations>

          ...  (repeat for all candidate messages)
        ]
```

Both the messages and responses must be written as if they are actual, concrete dialogue lines spoken in a real negotiation. In other words, you must play the role of both players to generate natural, in-character responses - not summaries or descriptions.

Each simulation must be fully completed - never stop midway. Simulate until the outcome is resolved for all 5 strategies.

Here is the list of candidate responses: {{concatenated_candidates}}

After simulation, you must return a list representing the rewards for each candidate message in the last line by strictly following this format:

```
        <reward list> [reward1, reward2, ...]  </reward list>
```

## B    ADDITIONAL EXPERIMENTAL RESULTS

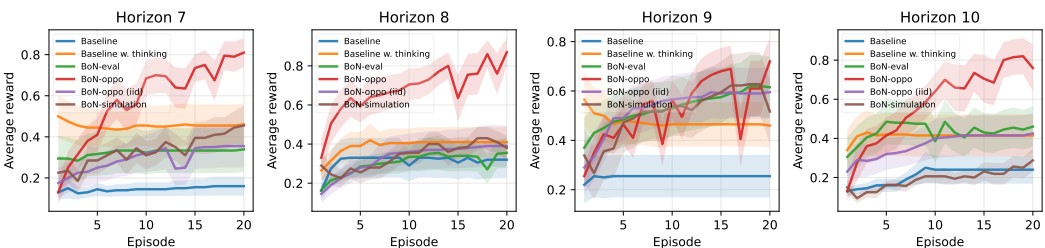

Figure 8: Seller's average rewards (normalized by 20) in games with different horizons.

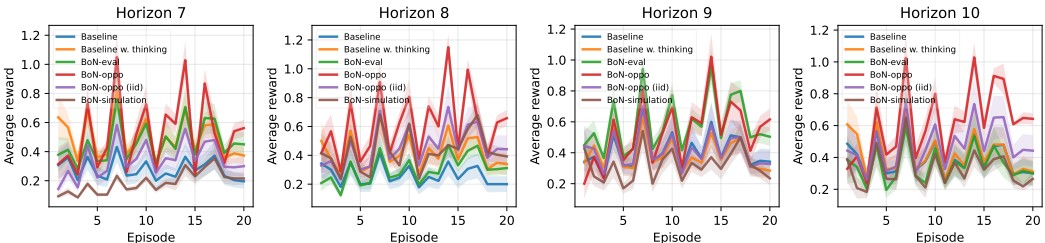

Figure 9: Seller's average rewards (normalized by the difference between the buyer's maximum willingness to pay and seller's production cost) in games where the buyer's maximum willingness to pay is uniformly sampled at the beginning of each episode.

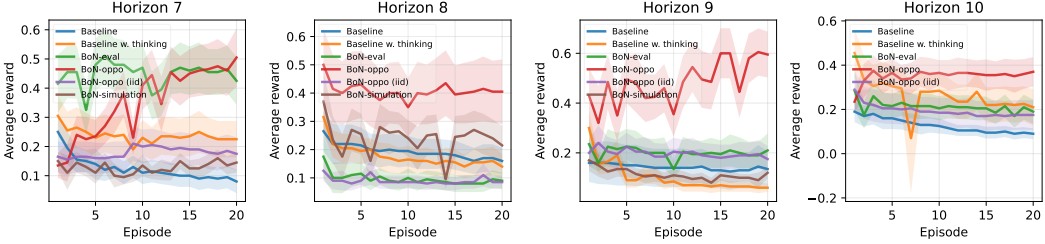

Figure 10: Seller's average rewards (normalized by 20) in games when competing against the buyer also adopting algorithm.

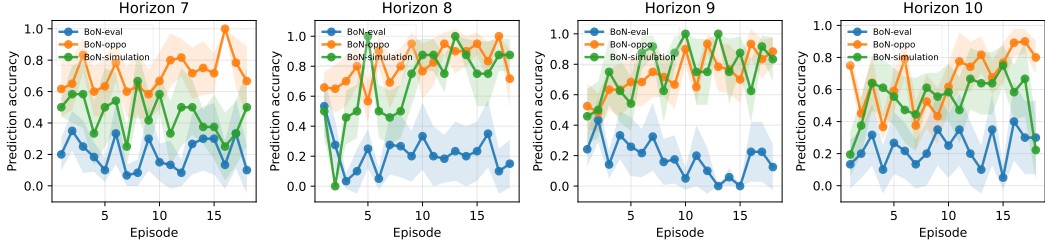

Figure 11: Buyer's accuracy of selecting the best candidate.

## C ADDITIONAL RELATED WORKS

**Opponent modeling in multi-agent RL.** Opponent modeling is a key technical component of our framework. Such techniques of opponent modeling have been an important ingredient of many successful (multi-agent) RL algorithms (He et al., 2016; Raileanu et al., 2018; Papoudakis et al., 2021; Yu et al., 2022; Weil et al., 2023), which introduce an auxiliary task of predicting the behavior of other agents from past interactions apart from the standard RL objective to address the infamous issues of non-stationarity. We refer to Albrecht & Stone (2018); Nashed & Zilberstein (2022) for a more comprehensive literature review. There is also another line of work explicitly accounting for the opponent for better stability and convergence of multi-agent learning dynamics (Foerster

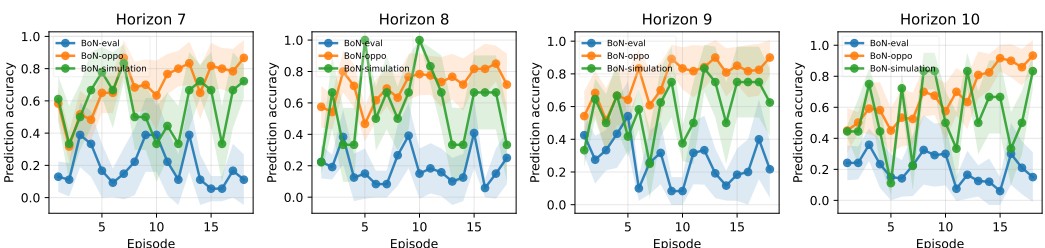

Figure 12: Seller's accuracy of selecting the best candidate.

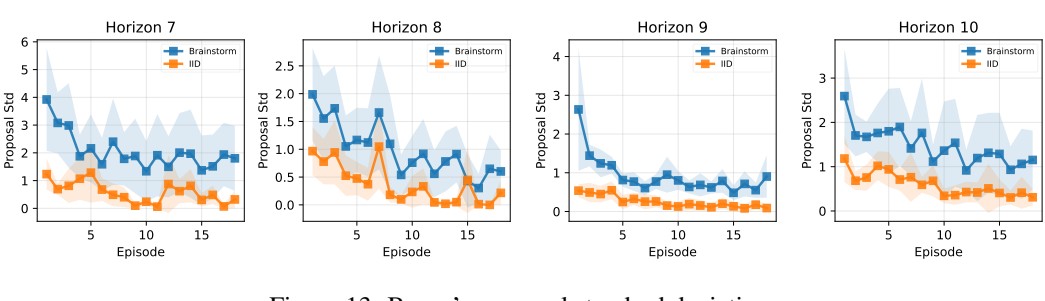

Figure 13: Buyer's proposal standard deviation.

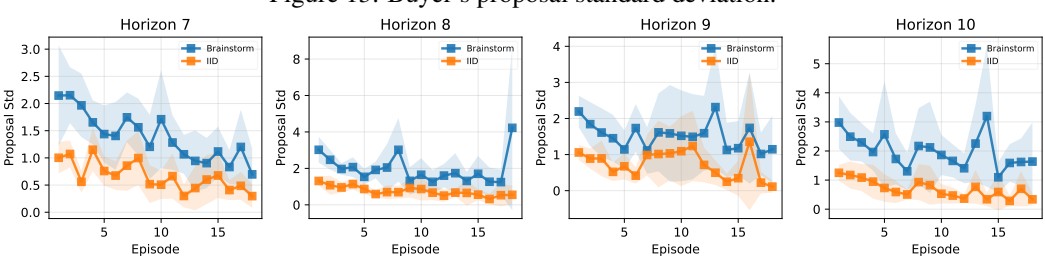

Figure 14: Seller's proposal standard deviation.

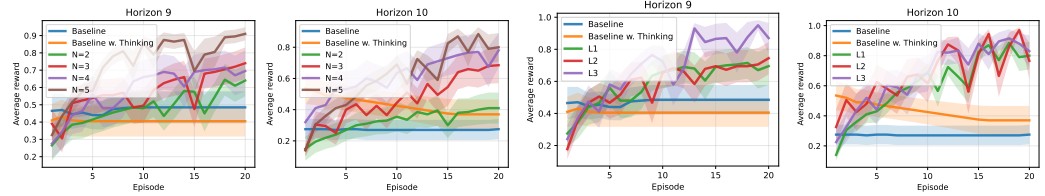

Figure 15: Results for scaling the number of candidates and higher-order BoN in the buyer-seller negotiation games.

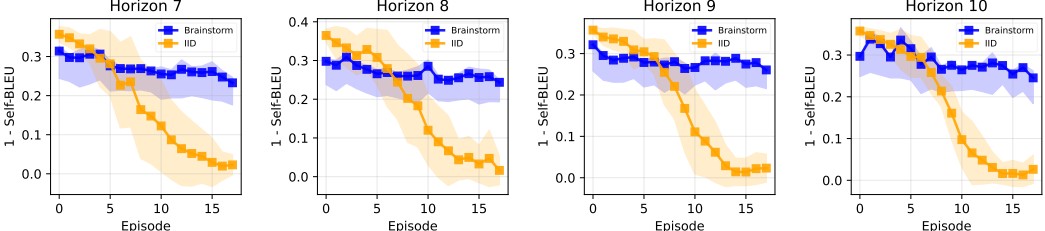

Figure 16: Diversity of buyer's candidate messages measured by $1-\mathrm{Self\_BLEU}$ (Shaib et al., 2024).

et al., 2018; Letcher et al., 2019; Lu et al., 2022). Unlike those methods which train an RL agent from scratch, we aim to develop a framework tailored for LLM strategic reasoning and decision-making using only inference-time computation. Recently, a closely related work Yu et al. (2025)

also explicitly models the opponents to predict their private information and feed such predictions to the acting agent to improve its decision-making ability instead of further exploiting the inference-time computation by using the opponent model as a *simulator* in our work.

**Inference-time techniques for LLM reasoning.** The success of OpenAI o1, Deepseek R1 has proven the effectiveness of the promising paradigm for LLMs reasoning by scaling the inference-time computation through prolonged thinking process (Snell et al., 2024; Welleck et al., 2024; Muennighoff et al., 2025). Apart from increasing a single thought trace, another effective way of scaling inference-time computation is by generating multiple candidates and choosing the best one, known as Best-of-$N$ sampling or parallel thinking (Google DeepMind, 2025; xAI, 2025). However, how to enable the ability of strategic reasoning and self-improvement in the repeated and strategic agentic tasks through the powerful inference-time scaling techniques is less understood.

# D DEFERRED PROOFS

## D.1 PROOF OF PROPOSITION 4.1

*Proof.* We start with the proof where the agent 1 takes the first turn. For any $\pi_1^\star \in \Pi_1$, we define the negotiation message that has the lowest probability as $\widehat{y}_{1,1}^m \in \arg\min_{y_{1,1}^m \in \mathcal{Y}_1^m} \sum_{y_{1,1}^p \in \mathcal{Y}_1^p} \pi_1^\star(y_{1,1}^p, y_{1,1}^m \mid x_1)$, where there is no history yet since it is the first turn. Now we construct an opponent policy $\pi_2$ that behaves as follows at the second step: if agent 2 receives the negotiation message $y_{1,1}^m = \widehat{y}_{1,1}^m$ and $y_{1,1}^p$ representing a proposal from the agent 1 that yields a non-negative reward for agent 2, it will immediately accept and ends the game. Otherwise, it will reject the proposal and end the game also. Now we define $r_1^{\max}$ as the maximum reward agent 1 can get subject to the constraint that agent 2's reward is non-negative. Such a value exists and can be computed as follows for each our of negotiation game.

- For the buyer-seller game, we have $r_1^{\max} = b - p$, where $b$ represents the buyer's maximum budget and $p$ represents the seller's production cost.

- For the resource exchange game, it is equivalent to solving the following program

$$r_1^{\max} = \max_{\Delta_X \in \mathbb{N}, \Delta_Y \in \mathbb{N}} v_1^X \cdot \Delta_X + v_1^Y \cdot \Delta_Y$$
$$\text{s.t. } v_2^X \cdot \Delta_X + v_2^Y \cdot \Delta_Y \le 0$$
$$\Delta_X \in [-n_1^X, n_2^X]$$
$$\Delta_Y \in [-n_1^Y, n_2^Y].$$

We denote the optimal solution as $\Delta_X^\star, \Delta_Y^\star$.

Therefore, by the construction of $\pi_2$, it holds that

$$V_1(\pi_1^\star, \pi_2) \le r_1^{\max} \cdot \mathbb{P}(y_{1,1}^m = \widehat{y}_{1,1}^m) \le \frac{r_1^{\max}}{|\mathcal{Y}_1^m|}.$$

Now we can construct the best response policy $\pi_1^\dagger$ against $\pi_2$ by letting $\pi_1^\dagger$ choose $(\widehat{y}_{1,1}^p, \widehat{y}_{1,1}^m)$ deterministically. $\widehat{y}_{1,1}^p$ simply chooses the proposal that maximizes agent 1's reward subject to the constraint that agent 2's reward is non-negative. Specifically,

- For the buyer-seller game, we set $\widehat{y}_1^p$ as the proposal of selling the product with price $b$ if agent 1 acts as the seller; otherwise, as the proposal of buying the product with price $p$ if agent 1 acts as the buyer.

- For the resource exchange game, we set $\widehat{y}_1^p$ as the proposal of getting $\Delta_X^\star$ of $X$ and $\Delta_Y^\star$ of $Y$ from agent 2. Note that if $\Delta_X^\star$ ($\Delta_Y^\star$) is negative, this means agent 1 gives $-\Delta_X^\star$ ($-\Delta_Y^\star$) of $X$ ($Y$) to agent 2.

By the construction of $\pi_1^\dagger$ and $\pi_2$, agent 2 will accept the proposal from the agent 1, yielding a reward of $r_1^{\max}$ for the agent 1. Formally, we have

$$\max_{\pi_1 \in \Pi_1} V_1(\pi_1, \pi_2) = V_1(\pi_1^\dagger, \pi_2) = r_1^{\max}.$$

This thus concludes that $V_1(\pi_1^\star, \pi_2) \leq \frac{\max_{\pi_1 \in \Pi_1} V_1(\pi_1, \pi_2)}{|\mathcal{Y}_1^m|}$.

For the case where agent 2 takes the first turn, for any given $\pi_1^\star \in \Pi_1$, we construct the policy $\pi_2$ similarly. At the first turn, agent 2 will deterministically choose $(y_{2,1}^p, y_{2,1}^m)$, where $y_{2,1}^p$ denotes waiting for a proposal, and $y_{2,1}^m$ denotes an empty string. Now we construct the policy $\pi_2$ at $h = 3$ by mimicking the construction of $\pi_2$ at $h = 2$ for the case above where the agent 1 takes the first turn. It is again straightforward to verify that $V_1(\pi_1^\star, \pi_2) \leq \frac{\max_{\pi_1 \in \Pi_1} V_1(\pi_1, \pi_2)}{|\mathcal{Y}_1^m|}$, thus concluding our proof. $\square$

### D.2 Proof of Proposition 4.2

*Proof.* We denote the action sequence played by the agent 2 as $b^{1:T}$. For each $t \in [T]$, we denote the reward vector $f^t := r_1(\cdot, b^t) \in \mathbb{R}^{|\mathcal{A}|}$. By the definition of $\pi_1^t$, for each $a \in \mathcal{A}$, we have

$$\pi_1^t(a) = \mathbb{P}\left(a \in \operatorname*{argmax}_{a' \in \mathcal{A}} \mathbb{E}_{b \sim \widehat{\pi}_2^t}[r_1(a', b)] + \eta_t \epsilon(a')\right)$$

$$= \mathbb{P}\left(a \in \operatorname*{argmax}_{a' \in \mathcal{A}} \frac{\sum_{t'=1}^{t-1} f^t(a')}{t-1} + \eta_t \epsilon(a')\right)$$

$$= \mathbb{P}\left(a \in \operatorname*{argmax}_{a' \in \mathcal{A}} \sum_{t'=1}^{t-1} f^t(a') + (t-1)\eta_t \epsilon(a')\right).$$

By Theorem 8 of (Abernethy et al., 2014), we have

$$\max_{\pi_1 \in \Delta(\mathcal{A})} \sum_{t=1}^T \left(\langle \pi_1, f^t \rangle - \langle \pi_1^t, f^t \rangle\right) \leq \sqrt{2 \log |\mathcal{A}|} \left((T-1)\eta^T + \sum_{t=1}^T \frac{\|f^t\|_\infty^2}{(t-1)\eta^t}\right).$$

Now by plugging in the choice of $\eta^t = \Theta(1/\sqrt{t})$, we conclude for any policy $\pi_1 \in \Delta(\mathcal{A})$

$$\sum_{t=1}^T \left(\langle \pi_1, f^t \rangle - \langle \pi_1^t, f^t \rangle\right) \leq \mathcal{O}(\sqrt{T \log |\mathcal{A}|}).$$

By taking expectations w.r.t. the random action sequences $b^{1:T}$ and noting that $\mathbb{E}_{b^t \sim \pi_2^t}[\langle \pi_1, f^t \rangle] = V_1(\pi_1, \pi_2^t), \mathbb{E}_{b^t \sim \pi_2^t}[\langle \pi_1^t, f^t \rangle] = V_1(\pi_1^t, \pi_2^t)$ for each $t \in [T]$, we conclude that

$$\mathbb{E}\left[\text{Regret}(T)\right] \leq \mathcal{O}(\sqrt{T \log |\mathcal{A}|}).$$

$\square$

### D.3 Proof of Theorem 4.4

*Proof.* We consider the case where agent 1 starts the first, i.e., $P(1) = 1$. The case where agent 2 starts the first can be proved similarly. We will prove by a backward induction on the time step $h$.

We firstly prove the base case. We denote $h^{\text{exit}} \in [H]$ the last time step agent 1 takes the action. If $H$ is an odd number, we have $h^{\text{exit}} = H$. In this case, we have for any $\tau_H^t$

$$V_{1,H}^{\pi_1^t, \pi_2^{\text{oppo}}}(\tau_H^t) = \mathbb{E}_{y_{1,H}^t \sim \pi_{1,H}^t(\cdot \mid \tau_H^t; \mathcal{C}^{t-1}, x^1)}\left[r_1(\tau_H^t, y_{1,H}^t)\right] = V_{1,H}^{\pi_1^t, \pi_2^t}(\tau_H^t).$$

Therefore, it holds that

$$\left|V_{1,H}^{\pi_1^t, \pi_2^{\text{oppo}}}(\tau_H^t) - V_{1,H}^{\pi_1^t, \pi_2^t}(\tau_H^t)\right| = 0$$

Meanwhile, if $H$ is an even number, we have $h^{\text{exit}} = H - 1$. In this case, we have for any $\tau_{H-1}^t$

$$\left| V_{1,H-1}^{\pi_1^t, \pi_2^{\text{oppo}}}(\tau_{H-1}^t) - V_{1,H-1}^{\pi_1^t, \pi_2^t}(\tau_{H-1}^t) \right|$$

$$= \left| \mathbb{E}_{y_{1,H-1}^t \sim \pi_{1,H-1}^t(\cdot \mid \tau_{H-1}^t; \mathcal{C}^{t-1}, x^1)} \mathbb{E}_{y_{2,H}^t \sim \pi_{2,H}^{\text{oppo}}(\cdot \mid (\tau_{H-1}^t, y_{1,H-1}^t); \mathcal{C}^{t-1})} \left[ r_1(\tau_{H-1}^t, y_{1,H-1}^t, y_{2,H}^t) \right] \right.$$

$$\left. - \mathbb{E}_{y_{1,H-1}^t \sim \pi_{1,H-1}^t(\cdot \mid \tau_{H-1}^t; \mathcal{C}^{t-1}, x^1)} \mathbb{E}_{y_{2,H}^t \sim \pi_{2,H}^t(\cdot \mid (\tau_{H-1}^t, y_{1,H-1}^t); \mathcal{C}^{t-1}, x^2)} \left[ r_1(\tau_{H-1}^t, y_{1,H-1}^t, y_{2,H}^t) \right] \right|$$

$$\leq \max_{y_{1,H-1}^t} d_{TV}\left( \pi_{2,H}^t(\cdot \mid (\tau_{H-1}^t, y_{1,H-1}^t); \mathcal{C}^{t-1}, x^2), \pi_{2,H}^{\text{oppo}}(\cdot \mid (\tau_{H-1}^t, y_{1,H-1}^t); \mathcal{C}^{t-1}) \right)$$

$$\leq \epsilon_H,$$

where the last step is by the definition of $\epsilon_H$.

Now we prove the case where $h < h^{\text{exit}}$ with $P(h) = 1$. Note that for any $\tau_h^t$, by Bellman equation, we have

$$\left| V_{1,h}^{\pi_1^t, \pi_2^{\text{oppo}}}(\tau_h^t) - V_{1,h}^{\pi_1^t, \pi_2^t}(\tau_h^t) \right|$$

$$= \left| \mathbb{E}_{y_{1,h}^t \sim \pi_{1,h}^t(\cdot \mid \tau_h^t; \mathcal{C}^{t-1}, x^1)} \mathbb{E}_{y_{2,h+1}^t \sim \pi_{2,h+1}^{\text{oppo}}(\cdot \mid (\tau_h^t, y_{1,h}^t); \mathcal{C}^{t-1})} \left[ V_{1,h+2}^{\pi_1^t, \pi_2^{\text{oppo}}}(\tau_h^t, y_{1,h}^t, y_{2,h+1}^t) \right] \right.$$

$$\left. - \mathbb{E}_{y_{1,h}^t \sim \pi_{1,h}^t(\cdot \mid \tau_h^t; \mathcal{C}^{t-1}, x^1)} \mathbb{E}_{y_{2,h+1}^t \sim \pi_{2,h+1}^t(\cdot \mid (\tau_h^t, y_{1,h}^t); \mathcal{C}^{t-1}, x^2)} \left[ V_{1,h+2}^{\pi_1^t, \pi_2^t}(\tau_h^t, y_{1,h}^t, y_{2,h+1}^t) \right] \right|$$

$$\leq \left| \mathbb{E}_{y_{1,h}^t \sim \pi_{1,h}^t(\cdot \mid \tau_h^t; \mathcal{C}^{t-1}, x^1)} \mathbb{E}_{y_{2,h+1}^t \sim \pi_{2,h+1}^{\text{oppo}}(\cdot \mid (\tau_h^t, y_{1,h}^t); \mathcal{C}^{t-1})} \left[ V_{1,h+2}^{\pi_1^t, \pi_2^t}(\tau_h^t, y_{1,h}^t, y_{2,h+1}^t) \right] \right.$$

$$\left. - \mathbb{E}_{y_{1,h}^t \sim \pi_{1,h}^t(\cdot \mid \tau_h^t; \mathcal{C}^{t-1}, x^1)} \mathbb{E}_{y_{2,h+1}^t \sim \pi_{2,h+1}^t(\cdot \mid (\tau_h^t, y_{1,h}^t); \mathcal{C}^{t-1}, x^2)} \left[ V_{1,h+2}^{\pi_1^t, \pi_2^t}(\tau_h^t, y_{1,h}^t, y_{2,h+1}^t) \right] \right|$$

$$+ (\epsilon_{h+3} + \epsilon_{h+5} + \cdots)$$

$$\leq \max_{y_{1,h}^t} d_{TV}\left( \pi_{2,h+1}^{\text{oppo}}(\cdot \mid (\tau_h^t, y_{1,h}^t); \mathcal{C}^{t-1}), \pi_{2,h+1}^t(\cdot \mid (\tau_h^t, y_{1,h}^t); \mathcal{C}^{t-1}, x^2) \right) + (\epsilon_{h+3} + \epsilon_{h+5} + \cdots)$$

$$\leq \epsilon_{h+1} + \epsilon_{h+3} + \cdots,$$

where we use the inductive hypothesis in the first inequality. Finally, by noting that

$$J_1(\pi_1^1, \pi_2^{\text{oppo}}) = V_{1,1}^{\pi_1^1, \pi_2^{\text{oppo}}}(\tau_1^h),$$

$$J_1(\pi_1^t, \pi_2^t) = V_{1,1}^{\pi_1^1, \pi_2^t}(\tau_1^h),$$

we proved the near optimality of the policy $\widehat{\pi}_1^t$ by the non-expansiveness of the max operator. $\qquad \square$

# E DISCUSSIONS AND IMPLEMENTATIONS OF ADDITIONAL BASELINES

Here we provide a detailed discussion on the three additional approaches from Fu et al. (2023), Xu et al. (2023), as well as Yu et al. (2025) considered in Section 5.

- **For Fu et al. (2023):** It introduces an additional critic at the beginning of each episode. The critic maintains all history and provides three (high-level) suggestions/feedbacks on how to improve the rewards in the next episode. Since the experimental setting resembles us, we can directly reuse its prompt in our implementations.

- **For Xu et al. (2023):** The primary goal of Xu et al. (2023) is to handle the issues of long contexts due to history accumulation in Werewolf games. Thanks to the recent advances of LLMs, long contexts are no long significant issues in our experiments. The core idea of Xu et al. (2023) is to retrieve one negative experience and several good experiences from the history. Then such experiences together with a short suggestion are fed to the acting agent at each decision-making step. Therefore, we call such approach selective experience reflection. Therefore, we mirror such implementation in our negotiation games and rank the decision in the entire negotiation history at each time step according to a score, which combines the final reward signal of that episode and a score from a critic.

---

**Algorithm 1** BoN-Opponent-Simulation (from the perspective of agent 1)

---

1: **Input:** $\pi_1^{\text{base}}, \pi_2^{\text{oppo}}, x_1, N, T, H$
2: **for** $t \in [T]$ **do**
3:     **for** $h \in [H]$ **do**
4:         **if** $P(h) = 1$ **then**
5:             **for** $k \in [N]$ **do**
6:                 Sample action $y_{1,h}^{t,k} \sim \pi_1^{\text{base}}(\cdot \mid \tau_h^t; \mathcal{C}^{t-1}, x_1)$
7:                 Simulate the episodes by first taking action $y_{1,h}^{t,k}$ and then following $(\pi_1^{\text{base}}, \pi_2^{\text{oppo}})$
    towards the end of the episode
8:                 Denote $\widehat{r}_1^k$ as the empirical average of the reward from the simulated trajectories
9:             **end for**
10:             $k^\star \leftarrow \text{argmax}_{k \in [N]} \widehat{r}_1^k$
11:             Take the action $y_{1,h}^{t,k^\star}$
12:             Update the partial trajectory $\tau_{h+1}^t \leftarrow (\tau_h^t, y_{1,h}^{t,k^\star})$
13:         **else**
14:             Observe the opponent action $y_{2,h}^t$
15:             Update the partial trajectory $\tau_{h+1}^t \leftarrow (\tau_h^t, y_{2,h}^t)$
16:         **end if**
17:     **end for**
18:     Update the context $\mathcal{C}^t \leftarrow (\mathcal{C}^{t-1}, \tau_{H+1}^t)$
19: **end for**

---

- **For Yu et al. (2025)**: It introduces an opponent model to predict the private information (specifically, player's role), in the WITU game. Then such private information, is also fed into the acting agent for better decision-making. To mirror such implementation, we let the opponent model predict the private information in our setting, i.e., production cost of the seller/budget of the buyer. Note that the opponent model in (Yu et al., 2025) is *not* used for simulation.

Finally, although we include comparisons with these approaches, we mainly regard them as evaluating the ability of our approach for handling adaptive opponents. Meanwhile, we remark the fundamental technical difference between our work and these related works: all the three works focus on how to provide better contexts/input prompts for the acting agent, *while the output of acting agent is kept native*. In contrast, we study how to *sharpen* the output distribution most effectively, while better context/prompt engineering is perpendicular to our main focus.

## F   Detailed description of our framework

In Algorithm 1, we describe the decision-making process using the perspective of the agent 1 for total $T$ episodes. At each episode $t \in [T]$, each time step $h \in [H]$, if it is agent 2's turn, i.e. $P(h) = 2$, agent 1 will observe the action $y_{2,h}^t$ from the opponent and update the partial trajectory. Otherwise, it will implement our BoN framework as in Section 4. Finally, we refer a graphical illustration of our framework to Figure 17.

## G   Example outputs of our agents

We refer the example outputs of our agents to the anonymous link https://github.com/llmnegotiationiclr-anonymous/llm-negotiation.

## H   The use of Large Language Models (LLMs) in paper writing

For the submission, we only use LLMs for proofreading purposes.

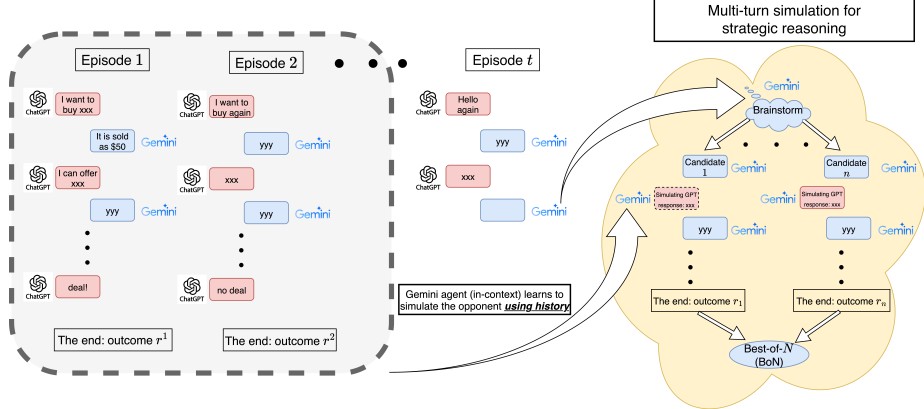

Figure 17: Graphical illustration of our framework.

| Model | Method | Buyer | Seller |
|-------|--------|-------|--------|
| **Gemini against Claude** | Baseline w. thinking | +0.63 ± 0.60 | +0.80 ± 3.75 |
| | BoN-eval | +1.03 ± 1.83 | -0.50 ± 2.44 |
| | BoN-simulation | +5.53 ± 3.45 | +1.03 ± 4.39 |
| | BoN-oppo (iid) | +3.23 ± 3.99 | -0.10 ± 4.29 |
| | **BoN-oppo** | **+6.94 ± 2.96** | **+4.86 ± 2.43** |
| **Gemini against Qwen** | Baseline w. thinking | +0.30 ± 0.46 | -0.57 ± 3.71 |
| | BoN-eval | +1.50 ± 1.50 | -1.61 ± 5.45 |
| | BoN-simulation | +2.17 ± 4.33 | +2.63 ± 6.67 |
| | BoN-oppo (iid) | +0.60 ± 1.02 | +0.10 ± 5.24 |
| | **BoN-oppo** | **+5.43 ± 3.57** | **+4.07 ± 2.26** |
| **Gemini against Llama** | Baseline w. thinking | -2.19 ± 6.51 | +1.73 ± 5.09 |
| | BoN-eval | +2.18 ± 3.50 | -0.81 ± 5.95 |
| | BoN-simulation | +5.41 ± 2.55 | +2.53 ± 7.20 |
| | BoN-oppo (iid) | +0.76 ± 7.07 | +3.50 ± 7.81 |
| | **BoN-oppo** | **+5.56 ± 2.57** | **+5.23 ± 2.60** |

Table 3: Results for our approach and baselines powered by Gemini playing against opponents powered by different base models. Bold indicates best average reward per model.

## I CONCLUDING REMARKS AND LIMITATIONS

In this paper, we demonstrate the potential of leveraging inference-time computation in strategic decision-making to enable continual self-improvement during repeated interactions. Several limitations are worth noting. First, our experiments primarily focus on two-agent negotiation settings; extending to multi-agent scenarios such as larger societies remains an important direction for future work. Second, motivated by the fictitious play dynamic, we deliberately design a generic opponent model to only approximate time-averaged behaviors *without making assumptions about the actual opponent*. However, in many real-world applications, it is often natural to assume access to some form of prior knowledge about the opponent. How to effectively embed such information into the opponent modeling framework is another promising avenue for future investigations.

## REPRODUCIBILITY STATEMENT

The core contribution of our paper is a framework for teaching LLMs in strategic reasoning and decision-making tasks. We have provided detailed descriptions of our idea in the main paper accompanied by pseudocode in Appendix F as well as concrete prompts for reproducing our experimental results

