# OpenReview forum: "Opponent Simulation as Inference-time Scaling for Self-improving Agent: Case Study of Repeated Negotiations"
_ICLR.cc/2026/Conference — Submitted to ICLR 2026_

### Official Review · Reviewer_oQJh · 2025-10-19

**Soundness:** 3
**Presentation:** 2
**Contribution:** 2
**Rating:** 4
**Confidence:** 4

**Summary:**

The paper proposes BoN-oppo-simulation, an inference-time framework for enabling LLMs to self-improve in repeated strategic interactions, such as negotiations, without offline training. Drawing from fictitious play (FP), it uses:

-An opponent model (separate LLM) that in-context learns time-averaged opponent behavior from history, with optimism under uncertainty for exploration.

-Best-of-N (BoN) sampling: Generate candidates via strategic brainstorming, simulate trajectories with the opponent model, and select the best based on rewards.

Evaluated on buyer-seller and resource exchange games, it shows self-improvement over episodes.

**Strengths:**

Embedding FP into LLM inference is somewhat novel, extending test-time compute scaling to multi-agent strategy. The opponent model as a simulator, with OFU, provides principled adaptation.

**Weaknesses:**

**Concept Confusion:** Prompt Engineering vs. Inference-Time Techniques: The paper positions its method as a "novel inference-time technique" (Sec. 2) superior to prior works like Fu et al. (2023), which it categorizes as "advanced technique for automatic prompt engineering." However, this distinction is unclear and potentially overstated. The proposed framework relies heavily on specialized prompts (e.g., for brainstorming strategies, reflection, opponent summarization in App. A)—which is essentially prompt engineering.

**Missing Baselines:** The related work   surveys LLM agents for strategic decision-making, citing Diplomacy agents (Bakhtin et al., 2022; Xu et al., 2025), opponent modeling (Yu et al., 2025), and others like Werewolf (Xu et al., 2023/2024). However, these are not empirically compared, despite their relevance to repeated, multi-agent strategy. A glaring omission is Richelieu (Guan et al., 2024) (Zhenyu Guan, Xiangyu Kong, Fangwei Zhong, and Yizhou Wang. Richelieu: Self-evolving llmbased agents for ai diplomacy. Advances in Neural Information Processing Systems, 37:123471–123497, 2024.), a self-evolving LLM agent for AI Diplomacy that uses hierarchical planning and self-play for adaptation—directly aligned with this paper's self-improvement goals. Richelieu achieves human-like performance without human data, making it a strong baseline for testing BoN-oppo-simulation in more complex games. Without these comparisons, the empirical claims feel insular, limited to toy negotiations against stationary opponents.

**Limited novelty**: Embedding FP into LLM inference has been studied by: Game of Thoughts: Iterative Reasoning in Game-Theoretic Domains with Large Language Models (AAMAS 2025)

**Limited Scope and Generalizability**: Evaluations are confined to two negotiation games. Why do we care about both games? How does this extend to more complex games (e.g., Diplomacy with partial observability, or continuous-action spaces like auctions)? Or non-negotiation settings (e.g., repeated Prisoner's Dilemma, bandits with LLM opponents)?

 Opponents are mostly stationary LLMs (e.g., Gemini without inference-time boosts). While they test dynamic budgets/costs, more adversarial or human opponents would strengthen claims. The paper acknowledges multi-agent limits in Sec. 6, but this feels like a key gap.

**Computational Overhead** is unclear.  BoN with simulations seems requiring high costs.

**Questions:**

See the above.

---

> ### Author Response · Authors · 2025-11-25
>
> ## To Reviewer oQJh
>
> We thank Reviewer oQJh for the insightful feedback and are encouraged that they find our approach **somewhat novel** and our adaptations **principled**. We address the concerns and questions below.
>
> ---
>
> > **Q1: Concept Confusion: Prompt Engineering vs. Inference-Time Techniques.**
> > The paper positions its method as a "novel inference-time technique" (Sec. 2) superior to prior works like Fu et al. (2023), which it categorizes as "advanced technique for automatic prompt engineering." However, this distinction is unclear and potentially overstated. The proposed framework relies heavily on specialized prompts (e.g., for brainstorming strategies, reflection, opponent summarization in App. A)—which is essentially prompt engineering.
>
> We believe there are some misunderstandings about our contribution claims and clarify them below.
>
> - **Conceptual differences between *input-level* methods like prompt/context engineering vs. *output-level* inference-time scaling.**
>   Following the categorization in recent work on scaling test-time compute [6], we explicitly distinguish two streams of methods:
>   - *Prompt engineering / in‑context learning*: modifying the **input** (instructions, exemplars, critics) to elicit better behavior, and then sampling a single trajectory.
>   - *Inference‑time scaling on the output side*: treating test‑time compute as a budget and explicitly searching over multiple **outputs** from the model (e.g., Best‑of‑\(N\), simulation, tree search).
>
>   Fu et al. [1] naturally falls into the first category: a critic rewrites the prompt at the beginning of each episode, and then the agent executes a single trajectory. Our BoN‑oppo‑simulation framework lies in the second category: at each decision, we generate multiple candidate actions and simulate their outcomes using an opponent model before choosing a best response. We refer to Snell et al. [6] for a more detailed discussion of these two styles (*input‑level* vs. *output‑level* methods). **Finally, we would like to clarify we have never claimed that one is superior than another. These are just distinct directions.**
>
> - **Relationship to Fu et al. [1].**
>   We do not intend to claim that Fu et al. [1] is less important. Rather, it explores a **distinct but equally important** research direction: automatically improving prompts and in‑context examples. Our experimental focus is on whether and how much **output‑side inference‑time interventions** (BoN + simulation) improve over other inference‑time‑based baselines, holding the model parameters fixed.
>
> - **On our use of prompt engineering.**
>   We do not claim to avoid prompt engineering. On the contrary, it is natural and necessary to use reasonably designed prompts for inference‑time approaches; no amount of compute can rescue a completely uninformative prompt. Our novelty does *not* lie in eliminating prompts, but in introducing a new **inference‑time algorithm** (implementing fictitious play with opponent modeling + BoN simulation) that enables self‑improvement in repeated negotiation.
>
> - **Clarification on “specialized prompts”.**
>   By “specialized prompts” we meant prompts that encode hand‑crafted heuristics or domain knowledge specific to the negotiation games (e.g., “increase the seller’s price by a fixed amount each episode” or explicit bluffing rules). We intentionally avoid such specialized heuristics because our focus is on understanding the effectiveness of (output‑level) inference‑time methods. As shown in Sec. A.2 and A.3, the prompts for **summarization, reflection, and opponent modeling** are deliberately kept generic: they ask the agent to summarize past rewards, reflect on what worked or failed, and simulate an opponent consistent with past behavior, but they *do not* encode hand‑crafted negotiation heuristics or domain knowledge about what a “good” negotiation strategy should be. For example, the reflection prompt simply asks the model to consider historical episodes and self‑improve, **without specifying *how* to do so**. We only refer to the specifics of the negotiation games when describing the game rules. To avoid confusion, we have revised the text to remove the word “specialized”.
>
> We hope this clarifies that our contribution lies in the **(output‑level) inference‑time scaling framework** (BoN‑oppo‑simulation) and its theoretical and empirical analysis in repeated negotiations, rather than in claiming that our prompts are simpler or that our approach is *superior* to prior work, which follows an orthogonal direction. Our intention is only to emphasize that Fu et al. [1] and our paper employ different, complementary technical styles and focus on distinct aspects of the problem.

---

> ### Author Response · Authors · 2025-11-25
>
> > **Q2a: Missing Baselines.**
> > The related work surveys LLM agents for strategic decision-making, citing Diplomacy agents (Bakhtin et al., 2022; Xu et al., 2025), opponent modeling (Yu et al., 2025), and others like Werewolf (Xu et al., 2023/2024). However, these are not empirically compared, despite their relevance to repeated, multi-agent strategy.
>
> - **Fundamentally different objective: equilibrium / less‑exploitable policy seeking vs. online no‑regret self‑improvement.**
>   Our objective is fundamentally different from that of these Diplomacy‑style agents. Those works train agents via large‑scale self‑play and fine‑tuning to output **one less‑exploitable policy**. In contrast, we tackle **continual self‑improvement during online interaction with unknown opponents**. Theoretically, this is not a property of a single policy, but of an adaptive policy sequence with ***no regret guarantees*** (as in our Proposition 4.2; see [12, 13] for this fundamental difference). Proposition 4.1 justifies this objective in our negotiation games: no single dominant policy can perform well against all opponents, which motivates an online‑learning / no‑regret viewpoint rather than searching for a single equilibrium policy.
>
> - **Regarding Bakhtin et al. (2022); Xu et al. (2025, 2024).**
>   We did not include these methods as baselines because they are based on pre‑training / fine‑tuning. Given our objective of online adaptation at test time, we believe such methods are less suitable because (i) they incur significantly higher latency and may struggle when opponents change frequently, and (ii) they require sufficient samples / interaction history to fine‑tune effectively. Therefore, we believe our inference‑time methods are better aligned with our test‑time problem setting. As Reviewer LZah kindly noted:
>   > The paper focuses on a critical real-world constraint: LLMs’ deployment in dynamic multi-agent settings where LLMs *cannot* rely on pre-trained/fine-tuned policies for unknown opponents.
>
> - **New experimental results on the remaining related works (Yu et al., 2025; Xu et al., 2023; Fu et al., 2023).**
>   These methods do not require parameter updates. We therefore include new experiments where our approach competes against opponents powered by each of them (*although some have different objectives, as clarified above*):
>   - **Opponents using the approach from Fu et al. [1] (AI feedback):** A critic maintains the full history and, at the beginning of each episode, provides high‑level suggestions on how to improve rewards in the next episode.
>   - **Opponents using the approach from Xu et al. [3] (selective experience reflection):** Following Xu et al. [3], the opponent retrieves one negative and several positive experiences from the history and conditions its next action on these retrieved examples plus a short suggestion; we refer to this as *selective experience reflection*.
>   - **Opponents using the approach from Yu et al. [4] (private information prediction):** Inspired by Yu et al. [4], the opponent maintains an explicit opponent model that predicts the other agent’s private information and feeds these predictions into its own prompt at each step.
>
>   For fair comparison, we symmetrize the buyer–seller game by letting all methods play both roles (buyer / seller) and start both first and second, and we normalize payoffs so that values \(> 0.5\) indicate outperforming the opponent. As shown in Figure 6, our BoN‑oppo‑simulation agent consistently achieves normalized rewards greater than 0.5 against all of these fast‑learning opponents, demonstrating robustness to a wide range of dynamic opponent behaviors.
>
> Therefore, we hope we have now properly discussed all related works mentioned by the reviewer and added all feasible experimental comparisons, and we are happy to make further revisions if helpful.
>
> ---

---

> ### Author Response · Authors · 2025-11-25
>
> > **Q2b: Richelieu (Guan et al., 2024) is a glaring omission.**
> > Richelieu is a self-evolving LLM agent for AI Diplomacy that uses hierarchical planning and self-play for adaptation—directly aligned with this paper's self-improvement goals. Richelieu achieves human-like performance without human data, making it a strong baseline for testing BoN-oppo-simulation in more complex games.
>
> - **Why Richelieu and Diplomacy agents are not used as direct baselines.**
>   We fully agree that Richelieu (Guan et al. [2]) is an important contribution to strategic LLM research. However, treating such systems as plug‑in baselines in our environment would be less relevant for several reasons:
>   1. **Training‑budget mismatch.** Richelieu evolves its agent via extensive self‑play and training, which implicitly assumes access to a large volume of simulated games. In our setting, we explicitly restrict online interaction to **10–20 episodes** against an unknown opponent, each with a single sparse reward. Under such a small interaction budget, it is non‑trivial for fine‑tuning‑based approaches to learn effectively, whereas inference‑time methods can already yield significant improvements, as we empirically demonstrated by our paper and literature.
>   2. **Orthogonal and complementary paradigms.** We view Richelieu‑style self‑evolving agents and our BoN‑oppo‑simulation framework as complementary rather than competing. One could in principle take a Diplomacy agent trained via self‑play and then apply our inference‑time FP / BoN framework on top to further adapt to a particular opponent. Conversely, one could distill our inference‑time behavior into a fine‑tuned model.
>
> - **Terminology: “self‑evolving” vs. “self‑improving”.**
>   We understand “self‑evolving” in Richelieu refers to training‑time self‑play **without human data**, with the goal of learning a less exploitable policy. In contrast, our notion of “self‑improving” refers to test‑time performance gains during online interaction against unknown opponents (cf. Remark 4.3 on the interpretation of self‑improvement in our setting). This terminology for test‑time self‑improvement is adopted from prior work on LLM negotiation with AI feedback [1]. We agree that terms such as “self‑evolving” and “self‑improving” are sometimes used with different objectives in the literature, and we clarify our usage accordingly here.
>
> ---
>
> > **Q2c: Without these comparisons, the empirical claims feel insular, limited to toy negotiations against stationary opponents.**
>
> - **Our opponents are not stationary.**
>   We would like to clarify that our opponents are never stationary in the strict sense:
>   1. The *default opponent LLMs* always maintain a memory of interactions from previous episodes. As history accumulates, their behavior naturally adapts and changes (as noted in Footnote 3).
>   2. We consider **opponents with dynamic budgets / costs**, where key parameters (e.g., seller’s production cost or buyer’s maximum willingness to pay) are re‑sampled at the beginning of each episode, inducing changing effective policies across episodes.
>   3. Most importantly, we evaluate against **opponents also using our inference‑time techniques**, i.e., both agents running BoN‑oppo‑simulation. This corresponds to a highly adaptive opponent that is also actively self‑improving based on historical feedback. Across horizons and both games, our method consistently outperforms all other baselines under this adaptive setting.
>
> - **New experimental results on more strongly adaptive opponents.**
>   In response to the reviewer’s suggestion, we further introduced **three new families of strongly adaptive opponents** from the literature, each powered by a distinct learning mechanism (AI feedback, selective experience reflection, and private‑information prediction; see Q2a/Q2b). We believe this further validates the ability of our approach to consistently self‑improve against and outperform even strongly adaptive opponents.
>
> **Nevertheless, we acknowledge that Richelieu and related Diplomacy agents have made significant contributions to strategic reasoning and decision‑making in complex games.** We have included all relevant citations of these works and view them as complementary to our (output‑level) inference‑time scaling framework. We are happy to make further revisions if the reviewer believes they would improve the paper.
>
> ---

---

> > ### Author Response · Authors · 2025-11-25
> >
> > > **Q3: Limited novelty: Embedding FP into LLM inference has been studied by Game of Thoughts: Iterative Reasoning in Game-Theoretic Domains with Large Language Models (AAMAS 2025).**
> >
> > We thank the reviewer for pointing out this related work. We agree that **Game of Thoughts (Kempinski et al. [5])** is highly relevant conceptually, but we believe it does not limit the novelty of our contributions for the following reasons.
> >
> > - **Fundamentally different problem focus (as in our response to Q1).**
> >   Game of Thoughts [5] focuses on improving **single‑shot decisions** in game‑theoretic domains (e.g., auctions, resource allocation) by letting an LLM iteratively refine its action through internal self‑play and level‑\(k\) / cognitive‑hierarchy‑style reasoning. Its goal is to find less exploitable actions in one‑shot games. In contrast, our emphasis is on **continual self‑improvement across repeated episodes against an external, unknown opponent**. We analyze this as a **no‑regret online‑learning problem** via smooth fictitious play (Proposition 4.2), rather than as a one‑shot equilibrium computation.
> >
> > - **Distinct utilization and implementation of game‑theoretic ideas.**
> >   While Game of Thoughts [5] draws inspiration from iterative best‑response reasoning, to our understanding it does *not* implement the smooth fictitious‑play dynamic as a learning process against the actual opponent. Instead, it runs internal self‑play where the LLM essentially plays all roles. Our work, by contrast:
> >   - maintains an explicit **opponent model** that in‑context learns to imitate the *real* opponent from past negotiation history (belief formation), and
> >   - uses this opponent model as a **simulator** to evaluate multiple candidate actions via BoN rollouts (best response).
> >
> > To conclude, while both our work and Game of Thoughts [5] use ideas inspired by game‑theoretic learning, they differ significantly in **problem focus, concrete algorithms, and experimental domains**. Our novelty does not lie in being the first to mention fictitious play in the LLM context, but in reviving the FP framework as a full (output-level) inference‑time scaling paradigm **for test‑time repeated LLM interactions with unknown external opponents** by mapping each of its conceptual components to practical LLM implementations and providing theoretical and empirical analysis.
> >
> > We appreciate the reviewer pointing out this paper and have included proper citations in the updated version.
> >
> > ---

---

> ### Author Response · Authors · 2025-11-25
>
> > **Q4: Limited Scope and Generalizability.**
> > Evaluations are confined to two negotiation games. Why do we care about both games? How does this extend to more complex games (e.g., Diplomacy with partial observability, or continuous-action spaces like auctions)? Or non-negotiation settings (e.g., repeated Prisoner's Dilemma, bandits with LLM opponents)? Opponents are mostly stationary LLMs (e.g., Gemini without inference-time boosts). While they test dynamic budgets/costs, more adversarial or human opponents would strengthen claims.
>
> - **Why these two negotiation games?**
>   The negotiation games we consider are **well‑accepted experimental environments** for understanding the strategic reasoning abilities of language models [7–9]. We chose the two games to cover complementary regimes: the buyer–seller game is more **competitive**, while the resource‑exchange game is more **cooperative and coordination‑heavy**. As we note in Sec. 3.1, this design allows us to evaluate whether our inference‑time scaling framework helps in both highly adversarial and highly cooperative settings.
>
> - **Positioning between Diplomacy and simple normal‑form games / bandits.**
>   Compared with Diplomacy:
>   1. Existing Diplomacy agents (including Richelieu) largely rely on **pre‑training / fine‑tuning with extensive self‑play**. As discussed in Q2, we believe such settings are challenging for any *purely* inference‑time scaling methods. However, this does not undermine the research value of studying such methods, as they do not contradict and often combined with pre‑training / fine‑tuning (e.g., high-profiled success of OpenAI o1/DeepSeek R1).
>   2. To the best of our knowledge, our work is the first to systematically study BoN‑based inference‑time scaling in repeated decision‑making. We therefore believe it is more reasonable to start from this “intermediate” negotiation setting rather than immediately jumping to full Diplomacy, which would require a major environment and training pipeline engineering efforts.
>
>   Compared with **repeated normal‑form games and bandits with LLM opponents**, our negotiation games are *richer* in that:
>   1. The action space is **unrestricted natural language**, not a small finite set of actions. Proposition 4.1 explicitly uses this structure to show that no dominant policy exists, highlighting the need for online adaptation.
>   2. The interaction is **multi‑turn**, with long dialogues and no intermediate numeric reward signals, which motivates our BoN‑simulation mechanism.
>   3. Agents must reason about **private information** and strategic communication, aspects that are absent or much simpler in canonical bandits or normal‑form games.
>
>   Overall, we view repeated natural‑language negotiation as lying between very complex environments like Diplomacy and simpler normal‑form games / bandits, and as particularly suitable for our goal of gaining scientific understanding of inference‑time scaling in strategic decision‑making.
>
> - **New experimental results against dynamic opponents.**
>   As discussed in Q2, our opponents are never purely stationary: they condition on the full interaction history, we re‑sample budgets / costs / preferences across episodes, and we consider opponents using our own inference‑time method as well as the three newly added families of strongly adaptive baselines (AI feedback, selective experience reflection, private‑information prediction). Under all these settings, our agent consistently self‑improves against the adaptive opponents.
>
> Overall, our goal is to introduce and analyze a **principled FP / BoN framework by scaling inference‑time computation**, not to claim general state‑of‑the‑art performance superior than all the other existing approaches (most of which focus on learning equilibrium policies via training).
>
> ---

---

> ### Author Response · Authors · 2025-11-25
>
> > **Q5: The paper acknowledges multi-agent limits in Sec. 6, but this feels like a key gap.**
>
> **Conceptually, our framework does not rely on there being exactly two agents.** Fictitious play naturally extends to multi‑player games, as documented in the classical game‑theory literature [10, 11], by treating *the rest of the population as a joint opponent*. The same idea applies here: in an $M$‑agent game, the BoN‑oppo‑simulation agent would use an opponent model that predicts the joint behavior of the $M-1$ other agents given the interaction history. **The BoN w. simulation framework itself is unchanged no matter whether there are $2$ agents or more.**
>
> The reason we believe scaling to many‑agent scenarios requires more effort is primarily on the engineering / implementation side of *one particular component*, the opponent model. There are several non‑trivial research questions about how to model other agents ***when there are many***. For example, there are multiple options:
> - (1) **Joint modeling.** Model the $M-1$ other agents as a single joint probability distribution. This is the most expressive representation but suffers from an exponential dependency on the number of agents, i.e., *the curse of multi‑agency*.
> - (2) **Individual modeling.** Model the other agents individually. This is highly scalable, as the complexity of the opponent model scales linearly with the number of agents, but it ignores potential correlations among opponents.
> - (3) **Structured modeling.** Interpolate between the above via a structured representation (e.g., an interaction graph) that captures key dependencies without modeling the full joint distribution.
>
> Therefore, we do not intend to overclaim that we have fully resolved the many-agent setting (whose opponent modeling remains a rich and active research topic in the multi-agent community) but rather aim to highlight these modeling questions as promising directions for future work.
> **Finally and most importantly, the focus of the paper is a principled framework of scaling inference‑time computation for self-improvement, whereas the opponent model is a flexible/customizable component.** We intentionally restrict ourselves to two‑agent negotiations in this work, since two‑player settings are already standard testbeds for strategic reasoning in LLMs [7–9] and allow us to cleanly evaluate inference‑time scaling ideas. Therefore, we believe studying how to extend opponent modeling to larger multi‑agent systems is a rich but less relevant direction, rather than a fundamental limitations of our central approach.
>
> ---
>
> > **Q6: Computational Overhead is unclear. BoN with simulations seems to require high costs.**
>
> In response to the reviewer’s suggestion, we added experiments and clarifications on computational overhead.
>
> - **Computation‑cost discussion.**
>   Theoretically, compared with the naive baseline without any inference‑time boosts, the ratio of computation costs between our BoN‑based methods and baseline methods scales **linearly** as
>   $$
>   \frac{\text{BoN costs}}{\text{baseline costs}} = \mathcal{O}(H \times N),
>   $$
>   where the dependence on $H$ (the number of time steps in one episode) arises because BoN‑based methods simulate each candidate over a (possibly complete) episode, and the dependence on $N$ arises because we need to evaluate $N$ candidates. **More importantly, for wall‑clock latency the dependence on $N$ can often be mitigated in practice**, since a major benefit of BoN‑based methods is that the $N$ candidates can be simulated in parallel given sufficient compute.
>
> - **New experimental results on token–performance trade‑offs.**
>   In Table 2, we report, for four different base models, the average performance boost over 20 repeated runs together with the $\log_2$ number of generated tokens for the baseline and our methods with different $N$. Using log‑scaled tokens follows standard practice in inference‑time scaling work. Table 2 shows that increasing inference‑time computation generally leads to higher performance, and that even relatively small $N$ (e.g., $N = 4$ or $6$) already achieves substantial gains across all base models, giving practitioners a tunable budget–performance trade‑off.
>
> - **Computation costs of typical inference‑time frameworks.**
>   We also clarify in Sec. 6 that existing inference‑time scaling methods (e.g., large‑depth chain‑of‑thought reasoning or extensive Best‑of‑$N$ sampling) often use inference‑time computation that is **an order of magnitude or more** higher than a single baseline decode. In this context, our BoN‑oppo‑simulation falls squarely within the usual test‑time compute budgets used in modern reasoning systems. **We therefore view the higher computation cost as an inherent and widely shared feature of inference‑time scaling approaches, rather than a limitation specific to our method.**

---

> > ### Author Response · Authors · 2025-11-25
> >
> > - **Efficiency relative to training‑time approaches.**
> >   Finally, we believe inference‑time scaling methods remain attractive because: (1) the additional inference‑time computation is generally much cheaper and more flexible than pre‑training or fine‑tuning when adapting to new opponents or domains; and (2) as our experiments show, they can be **highly sample‑efficient**, requiring only 10–20 episodes to obtain significant self‑improvement.
> >
> > We greatly appreciate Reviewer oQJh’s valuable feedback and constructive suggestions. We have incorporated all feasible experiments suggested by the reviewer. We hope our responses address the questions and concerns, and we are happy to make additional revisions or clarifications to further improve the paper.
> >
> > *Authors*
> >
> > ---
> >
> > ## References
> >
> > [1] Fu, Yao, et al. “Improving language model negotiation with self‑play and in‑context learning from AI feedback.” arXiv preprint arXiv:2305.10142, 2023.
> >
> > [2] Guan, Zhenyu, Xiangyu Kong, Fangwei Zhong, and Yizhou Wang. “Richelieu: Self‑evolving LLM‑based agents for AI Diplomacy.” *Advances in Neural Information Processing Systems* 37 (2024): 123471–123497.
> >
> > [3] Xu, Yuzhuang, et al. “Exploring large language models for communication games: An empirical study on Werewolf.” arXiv preprint arXiv:2309.04658, 2023.
> >
> > [4] Yu, XiaoPeng, Wanpeng Zhang, and Zongqing Lu. “LLM‑based explicit models of opponents for multi‑agent games.” In *Proceedings of the 2025 Conference of the Nations of the Americas Chapter of the Association for Computational Linguistics: Human Language Technologies (Volume 1: Long Papers)*, 2025.
> >
> > [5] Kempinski, Benjamin, et al. “Game of Thoughts: Iterative reasoning in game‑theoretic domains with large language models.” In *Proceedings of AAMAS*, 2025.
> >
> > [6] Snell, Charlie, et al. “Scaling LLM test‑time compute optimally can be more effective than scaling model parameters.” arXiv preprint arXiv:2408.03314, 2024.
> >
> > [7] Bianchi, Federico, et al. “How well can LLMs negotiate? NegotiationArena platform and analysis.” In *Proceedings of ICML*, 2024.
> >
> > [8] Lewis, Mike, Denis Yarats, Yann Dauphin, Devi Parikh, and Dhruv Batra. “Deal or No Deal? End‑to‑End Learning of Negotiation Dialogues.” In *Proceedings of the 2017 Conference on Empirical Methods in Natural Language Processing*, pp. 2443–2453, Copenhagen, Denmark. Association for Computational Linguistics, 2017.
> >
> > [9] Davidson, Tim Ruben, Veniamin Veselovsky, Michal Kosinski, and Robert West. “Evaluating language model agency through negotiations.” In *Proceedings of ICLR*, 2024.
> >
> > [10] Monderer, Dov, and Lloyd S. Shapley. “Fictitious play property for games with identical interests.” *Journal of Economic Theory* 68.1 (1996): 258–265.
> >
> > [11] Sela, Aner. “Fictitious play in ‘one‑against‑all’ multi‑player games.” *Economic Theory* 14.3 (1999): 635–651.
> >
> > [12] Shalev‑Shwartz, Shai. “Online learning and online convex optimization.” *Foundations and Trends in Machine Learning* 4.2 (2012): 107–194.
> >
> > [13] Cesa‑Bianchi, Nicolo, and Gábor Lugosi. *Prediction, Learning, and Games.* Cambridge University Press, 2006.

---

> > > ### Comment · Reviewer_oQJh · 2025-11-26
> > >
> > > I encourage authors to includes more baselines and tasks in experiments.

---

> ### Author Response · Authors · 2025-11-26
> **Thanks for your response**
>
> We thank the reviewer for the follow-up comment and apologize if our previous rebuttal did not sufficiently clarify our choices of baselines and tasks.
>
>
> - **We have carefully enumerated all related papers mentioned in the review, included all citations in the revised paper, and added all relevant baselines.** Among these works (Bakhtin et al., 2022; Xu et al., 2025; Fu et al., 2025; Xu et al., 2023; Xu et al., 2024; Yu et al., 2025; Guan et al., 2024), we have incorporated all baselines that do not rely on fine-tuning. We would like to gently reiterate our point from the previous response: fine-tuning-based methods pursue different goals and are fundamentally unsuitable for our inherently test-time setting. *This point was indeed also explicitly raised and emphasized by Reviewer LZah.* However, we are also open to any further ways of contrasting/clarifying/acknowledging the differences with these fine-tuning-based works.
>
>
> - **Negotiations (and simpler settings) are also used as the sole experimental setting in certain related work mentioned by the reviewer.** For instance, Fu et al. (2023) in fact only considers one of our experimental settings, namely the buyer–seller game. Kempinski et al. (2025) focuses exclusively on the simpler normal-form games (which is single-turn). Other related works similarly limit their attention to their own specific experimental setting. In our case, we have explicitly stated in the title that our work targets the negotiation setting, which has been extensively adopted in the literature (e.g., Lewis et al., 2017; Davidson et al., 2024; Bianchi et al., 2024), rather than being a toy problem invented by ourselves.
>
>
> **That said, we are more than happy to further revise our paper to include any additional discussions or comparisons with any other related works the reviewer has in mind and could kindly point out specifically.** We would also be extremely grateful if the reviewer could also take the objective of our paper and its technical focus into account when assessing paper. We would like sincerely thank the reviewer's engagement in advance again!

---

### Official Review · Reviewer_oqQo · 2025-10-31

**Soundness:** 3
**Presentation:** 3
**Contribution:** 3
**Rating:** 6
**Confidence:** 2

**Summary:**

This paper proposes the BoN-oppo-simulation framework, which facilitates adversarial negotiation reasoning for LLMs at inference time. The interactions between LLM agents are modeled through the lens of fictitious play. A case study in repeated negotiation games demonstrates that the proposed method achieves significant self-improvement over repeated interaction compared with various baselines.

**Strengths:**

- The paper is presented with exemplary clarity, greatly aided by its well-defined and intuitive game setting.

- It makes a significant and pragmatic contribution by introducing a tractable, inference-time LLM framework for competitive negotiation scenarios.

- The methodology is comprehensive and well-grounded, and I believe it to be reproducible, as evidenced by the provided example outputs.

- The paper presents a highly interesting and original integration of game-theoretic reasoning into LLMs. This integration is supported by substantial theoretical justification.

**Weaknesses:**

- The presentation of the game-theoretic model could be enhanced by incorporating a visual overview, such as a diagram, to illustrate the overall framework and interaction flow.

- In Section 3.1, the description of two games could be better grounded in the LLM context, clarifying how their designs specifically leverage or challenge LLM reasoning capabilities.

**Questions:**

- On Page 1, the minimax strategy appears overly conservative in scenarios that are not highly adversarial. Could the authors clarify what level or degree of adversarial setting this work specifically targets?

- Traditional adversarial game theory offers a variety of well-established models. Why do the authors choose to model the problem via fictitious play, rather than adopting other well-established game formulations?

- When multiple LLMs engage in repeated games as described in this paper, is there a risk of uncontrolled amplification (e.g., hallucination) emerging during the interaction process?

---

> ### Author Response · Authors · 2025-11-25
>
> ## To Reviewer oqQo
> We thank Reviewer oqQo for the insightful feedback. We are encouraged that Reviewer oqQo finds that our contributions are **significant and pragmatic**, our methodology is **comprehensive and well-grounded**, and our integrations of strategic reasoning in LLMs are **highly interesting and original**. We here address Reviewer oqQo's concerns and questions below:
>
> > Q1: The presentation of the game-theoretic model could be enhanced by incorporating a visual overview, such as a diagram, to illustrate the overall framework and interaction flow.
>
> We thank the reviewer’s kind suggestion and have incorporated a visual overview of our framework in the revised paper. In particular, we refer the reviewer to **Figure 17** in Appendix F, which provides a graphical illustration of our framework. The diagram summarizes the overall interaction flow across episodes, including (i) candidate generation via best-of-N sampling, (ii) opponent modeling and multi-step simulation, and (iii) the update of the shared negotiation history that both the acting agent and opponent model condition on. We believe this figure makes the game-theoretic structure and inference-time workflow much clearer.
>
> > Q2: In Section 3.1, the description of two games could be better grounded in the LLM context, clarifying how their designs specifically leverage or challenge LLM reasoning capabilities.
>
> We appreciate the reviewer’s feedback on improving the clarity of the problem setup. In the revised Section 3.1, we have added additional explanation of why these negotiation games are particularly interesting for LLM research. In particular, we now explicitly highlight that (i) the action space is in natural language rather than discrete symbolic actions, (ii) both games involve private information and require reasoning over accumulated interaction history, and (iii) the two games span different mixtures of competition and cooperation, thereby stressing different aspects of strategic reasoning. Meanwhile, it is worth pointing out that negotiation games are already a well-motivated experimental environment for understanding LLM strategic reasoning and decision-making, as discussed in prior work [1, 2, 3]. To the best of our knowledge, we are the first to utilize such games specifically to study **inference-time interventions** for strategic reasoning and decision-making.
>
> > Q3: On Page 1, the minimax strategy appears overly conservative in scenarios that are not highly adversarial. Could the authors clarify what level or degree of adversarial setting this work specifically targets?
>
> We thank the reviewer for touching on the key motivation of our paper. The very core motivation of our framework is that LLM agents should have the ability to self-improve during online interaction with an *unknown* opponent. Our framework does not rely on prior knowledge of the opponent (such as the degree of adversariality); instead, the agent continually collects information from the interaction history and forms beliefs about the opponent’s behavior from it.
>
> Regarding the reviewer’s specific question: we target **mixed-motive repeated interactions** where the environment is neither purely zero-sum nor fully cooperative, and where the opponent may be unknown and potentially dynamic. In such settings, offline minimax-style strategies can indeed be overly conservative when the actual opponent is not worst-case adversarial. For our two negotiation games, Proposition 4.1 formally shows that there is no single dominant policy that performs optimally against all possible opponent behaviors; in fact, any fixed strategy can be made arbitrarily bad by an appropriately chosen opponent. This motivates our choice to focus on inference-time self-improvement rather than pre-training a single fixed policy.
>
> Finally, we have intentionally chosen our two negotiation games to span a broad range of adversariality: the buyer–seller game is highly competitive (but not zero-sum), while the resource-exchange game requires a substantial amount of coordination (but is not fully cooperative). We believe these two settings together provide a realistic and challenging spectrum of opponent behaviors beyond the extremes for which minimax is designed.

---

> > ### Author Response · Authors · 2025-11-25
> >
> > > Q4: Traditional adversarial game theory offers a variety of well-established models. Why do the authors choose to model the problem via fictitious play, rather than adopting other well-established game formulations?
> >
> > We thank the reviewer for this insightful question and clarify our approach as follows.
> >
> > - **Fictitious play directly supports online adaptation with provable guarantees.** Arguably, many traditional game formulations and solution concepts (e.g., minimax in zero-sum games or Nash equilibrium computation) focus on offline equilibrium computation and are thus less suitable for our goal of continual self-improvement during online interaction. In contrast, fictitious play (more precisely, its smooth variant) can not only serve as an equilibrium-computation routine in classical settings, but also enjoys provable *no-regret guarantees* as shown in Proposition 4.2. This means that, when interpreted through its equivalence to Follow-The-Perturbed-Leader, the fictitious-play dynamic guarantees that the learner’s long-run performance approaches that of the best fixed policy in hindsight, against **arbitrary sequences of opponent policies**. On the other hand, most game-theoretic solvers—traditional ones like linear programming for equilibria computation, or modern multi-agent RL-based methods such as PSRO [4] are not primarily designed as online learning algorithms with such explicit no-regret guarantees in this sense.
> >
> > - **It is conceptually simple, general, and compatible with LLM implementations.** As we demonstrate in the paper, the fictitious-play perspective naturally decomposes our framework into two relatively independent components: (i) **belief formation** via in-context opponent modeling, and (ii) **best response computation** via BoN. This decomposition is conceptually clean and aligns well with the strengths of LLMs: belief formation can leverage progress in opponent modeling and summarization, while best response computation can benefit from advances in inference-time scaling and strategic candidate generation. We believe both components are of great research value on their own and may benefit from future progress in related LLM research areas.
> >
> > > Q5: When multiple LLMs engage in repeated games as described in this paper, is there a risk of uncontrolled amplification (e.g., hallucination) emerging during the interaction process?
> >
> > We really appreciate the reviewer for highlighting this potential risk. In our current experiments, we have not observed any signs of uncontrolled amplification or runaway behavior in the negotiation transcripts (see the example outputs provided in the anonymous repository at https://github.com/llmnegotiationiclr-anonymous/llm-negotiation). We believe this is largely attributable to the extensive safety alignment and guardrails present in the high-profile models used in our paper, combined with the relatively constrained nature of our environments (structured negotiation tasks with clear goals and rules).
> >
> > At the same time, we agree that as LLM agents are deployed in more open-ended and less constrained multi-agent environments, understanding and regulating the dynamics of fully automated LLM–LLM interactions will become an important research topic. We see our work as a step toward principled inference-time control in such settings, rather than a complete solution to the broader safety challenges.
> >
> > We greatly appreciate Reviewer oqQo’s valuable feedback and constructive suggestions. We hope our responses have addressed all the questions and concerns the reviewer had. We are happy to answer any further questions.
> >
> > Paper6002 Authors
> >
> >
> > ---
> >
> > References:
> >
> > [1] Mike Lewis, Denis Yarats, Yann Dauphin, Devi Parikh, and Dhruv Batra. 2017. Deal or No Deal? End-to-End Learning of Negotiation Dialogues. In Proceedings of the 2017 Conference on Empirical Methods in Natural Language Processing, pages 2443–2453, Copenhagen, Denmark. Association for Computational Linguistics.
> >
> >
> > [2] Federico Bianchi, Patrick John Chia, Mert Yuksekgonul, Jacopo Tagliabue, Dan Jurafsky, and James Zou. “How well can LLMs negotiate? NegotiationArena platform and analysis.” *ICML*, 2024.
> >
> > [3] Tim Ruben Davidson, Veniamin Veselovsky, Michal Kosinski, and Robert West. “Evaluating language model agency through negotiations.” *ICLR*, 2024.
> >
> > [4] Lanctot, Marc, et al. "A unified game-theoretic approach to multiagent reinforcement learning." Advances in neural information processing systems 30 (2017).

---

### Official Review · Reviewer_LZah · 2025-10-31

**Soundness:** 2
**Presentation:** 3
**Contribution:** 2
**Rating:** 4
**Confidence:** 2

**Summary:**

This paper focuses on the limitation of LLMs in achieving continual self-improvement during repeated strategic interactions with no prior knowledge about the opponents. To tackle this problem, it proposes BoN-oppo-simulation, a lightweight inference-time framework that embeds the game-theoretic principle of fictitious play (FP) into LLM implementations. The framework uses a separate LLM as an opponent model to learn and imitate the opponent’s time-averaged behavior from past interactions for belief formation, and adopts structured strategic brainstorming to generate candidate responses, then selects the optimal one via opponent-simulated future outcomes for best response. Empirical experiments on buyer-seller and resource exchange negotiation games across multiple LLMs demonstrate that the framework outperforms the baselines, motivating the necessity of self-improvement for repeated strategic decision-making.

**Strengths:**

1. The paper introduces a novel BoN-oppo-simulation framework. It lies in the non-trivial fusion of classical fictitious play (FP) in game theory with LLM inference-time scaling, that it embeds FP’s core steps (belief formation, best response) into practical LLM pipelines, enabling continual self-improvement via interaction feedback without fine-tuning for repeated strategic decision-making.
2. This work provides some theoretical analysis to justify the necessity of self-improvement and the "no-regret" property of the FP-based design. And it also empirically tests the proposed framework with granular implementation details across LLMs and negotiation games to demonstrate the efficacy.
3.  The paper focuses on a critical real-world constraint: LLMs’ deployment in dynamic multi-agent settings where LLMs cannot rely on pre-trained/fine-tuned policies for unknown opponents. This paper may provide a practical solution for real-world strategic LLMs.

**Weaknesses:**

1. The paper frames the framework as lightweight and scalable. However, it provides insufficient quantitative analysis of inference-time computational cost compared with baselines. For example, BoN sampling with N=5 and multi-turn opponent simulation may incur higher costs, but the paper does not discuss tradeoffs between performance and computational efficiency, which is critical for real-world deployment.

2.  The paper states that the proposed framework is a general inference-time framework. However, i) it lacks analysis of opponent model generalization across LLM architectures, that more details on how the opponent model’s performance varies with different base models are needed; ii) the dynamic opponents tested in this paper are relatively simple, and the paper does not evaluate performance against highly adaptive opponents. These may limit the confidence in the framework’s robustness to adversarial dynamic strategies.

3. The paper claims structured brainstorming generates more diverse candidates than i.i.d. sampling but relies solely on standard deviation as a diversity metric.  I think it is hard to confirm if brainstorming truly explores more strategic space or only varies numerical values. In natural language negotiations, maybe more dimensions and diversity metrics, e.g., measuring semantic similarity, should be considered for the quantitative comparison of the candidate generations.

4. The number of candidates N is set to 5 by default. However, the paper lacks investigation of how hyperparameters, e.g., N, impact the performance and the computational cost.  The analysis of hyperparameter sensitivity should be considered for practical deployment, as users need to know how to tune N for their resource constraints.

**Questions:**

Please refer to weaknesses.

---

> ### Author Response · Authors · 2025-11-25
>
> ## To Reviewer LZah
> We thank Reviewer LZah for the insightful feedback. We are encouraged that Reviewer LZah finds that our contributions are **novel and nontrivial**, and we are excited that the reviewer acknowledges that **our paper may provide a practical solution for real-world strategic LLMs**. We here address Reviewer LZah's concerns and questions below:
>
> > Q1: The paper frames the framework as lightweight and scalable. However, it provides insufficient quantitative analysis of inference-time computational cost compared with baselines. For example, BoN sampling with N=5 and multi-turn opponent simulation may incur higher costs, but the paper does not discuss tradeoffs between performance and computational efficiency, which is critical for real-world deployment.
> >
> We thank the reviewer for highlighting this important point regarding deployment.
>
> - ***Discussions on the computation costs:*** Theoretically speaking, compared with the naive baseline without any inference-time boosts, the ratio of computation costs between our BoN-based methods and baseline methods scales **linearly** as
> $$
> \frac{\text{BoN Costs}}{\text{Baseline Costs}} = \mathcal{O}(H \times N),
> $$
> where the dependency on $H$ (the number of time steps in one episode) arises because BoN-based methods simulate each candidate over a (possibly complete) episode, and the dependency on $N$ arises because we need to evaluate $N$ candidates. **More importantly, for wall-clock latency, the dependency on $N$ can be largely mitigated in practice, since one major benefit of BoN-based methods is that the $N$ candidates can be simulated in parallel given sufficient compute.**
>
> - ***New experimental results on tokens vs. performance trade-off:*** In our we revised paper, we added Table 2, which reports, for four different base models, the average performance boost over 20 repeated runs together with the (log$_2$) number of generated tokens for the baseline and our methods with different $N$. Using the log scale for token counts follows standard practice in inference-time scaling works (e.g., [1, 2]). We see in Table 2 that increasing inference-time computation generally leads to higher performance, and that even relatively small $N$ (e.g., $N=4$ or $6$) already achieves substantial gains across all base models, giving practitioners a tunable budget–performance trade-off.
>
> - **Remarks on the computation costs of typical inference-time framework in the literature:** Finally, we would like to remind the reviewer that existing inference-time scaling methods often use inference-time computation that are orders of magnitude higher than the baseline, e.g., [1, 2]. **Therefore, we view the higher computation cost as an inherent and widely common cost of inference-time scaling approaches, rather than a limitation specific to our method.**

---

> ### Author Response · Authors · 2025-11-25
>
> > Q2: The paper states that the proposed framework is a general inference-time framework. However, i) it lacks analysis of opponent model generalization across LLM architectures, that more details on how the opponent model’s performance varies with different base models are needed; ii) the dynamic opponents tested in this paper are relatively simple, and the paper does not evaluate performance against highly adaptive opponents. These may limit the confidence in the framework’s robustness to adversarial dynamic strategies.
>
> - **For analysis on opponent model generalization:** We would like to remind the reviewer that, in our main results (Table 1), we already vary the architecture of the acting agent and the opponent model (Gemini, Claude, Qwen, LLaMA) while keeping the *true opponent* fixed to Gemini. Therefore, in these settings, the opponent model is often architecturally different from the real opponent it is modeling, yet Table 1 shows that our method can still reliably achieve significant performance increases compared with baselines. This directly reflects the opponent model’s ability to generalize across LLM architectures.
>
> - **New experimental results on opponent architecture generalization:** In response to the reviewer’s suggestion, we additionally include experiments with opponents powered by Claude, Qwen, and LLaMA in Table 3. In these experiments, our method (powered by Gemini) plays against opponents with different backbone architectures. We can see that even in these cross-architecture settings, our method still achieves the most significant performance gains relative to baselines.
>
>
> - **For dynamic opponents:** Firstly, we would like to point out that there are already several forms of dynamic behavior across our experiments: (1) The *default opponent LLMs* maintain a memory of interactions from historical episodes and therefore naturally adapt/change their behaviors as the history accumulates (as noted in our footnote 3). (2) We consider **opponents with dynamic budgets/costs**, where key parameters are re-sampled at the beginning of each episode. (3) Most importantly, we evaluate against **opponents also using our inference-time techniques**, i.e., both agents running BoN-oppo-simulation. **We believe this already corresponds to a highly adaptive opponent actively self-improving based on historical feedback, and our method still consistently outperforms all other baselines in this setting.**
>
> - **New experimental results on more strongly adaptive opponents:** In response to the reviewer’s suggestions, we introduce three families of strongly adaptive opponents, each powered by a distinct learning mechanism:
>
>   - **Opponents using the approach from [3] (AI feedback):** This method introduces an additional critic at the beginning of each episode. The critic maintains all history and provides three high-level suggestions/feedbacks on how to improve rewards in the next episode.
>
>   - **Opponents using the approach from [4] (selective experience reflection):** The core idea of [4] is to retrieve one negative experience and several good experiences from the history, and then feed these retrieved examples plus a short suggestion to the acting agent at each decision-making step. We therefore refer to this approach as *selective experience reflection*.
>
>   - **Opponents using the approach from [5] (private information prediction):** This method introduces an opponent model that predicts the opponent’s private information, which is then fed into the acting agent to support better decision-making.
>
> We report the results in Figure 6, where we symmetrize the buyer-seller game by letting all methods play both roles (buyer/seller) and start both first and second, and then normalize the rewards so that a value \(> 0.5\) indicates outperforming the opponent. Under this evaluation, our method consistently outperforms these fast-learning opponents. We refer the reviewer to Section 5 for the revised discussion of dynamic opponents, and to Section E for more detailed descriptions of these three additional approaches.

---

> ### Author Response · Authors · 2025-11-25
>
> > Q3: The paper claims structured brainstorming generates more diverse candidates than i.i.d. sampling but relies solely on standard deviation as a diversity metric. I think it is hard to confirm if brainstorming truly explores more strategic space or only varies numerical values. In natural language negotiations, maybe more dimensions and diversity metrics, e.g., measuring semantic similarity, should be considered for the quantitative comparison of the candidate generations.
>
> We thank the reviewer for pointing out the limitation of our original diversity metrics. As per the reviewer's suggestion, we additionally measure ***semantic similarity*** and diversity measured by Self-BLEU [6] in Figure 5 and 6. Specifically, we compute the diversity score for a set of candidate responses in the following two ways:
> - Semantic diversity:
> $$
> \text{Diversity}(\{e_i\}\_{i\in[N]}) = 1 - \frac{\sum\_{i\in[N]}\sum\_{j\in[N]: j\neq i} \arccos(\langle e_i, e_j\rangle)}{N(N-1)},
> $$
> where each $e_i$ denotes the embedding (from Gemini embedding models) of candidate response $i$. This metric measures the average pairwise angular distance between candidate embeddings, so higher values indicate that candidates are more semantically spread out.
> - Self-BLEU: another commonly adopted text diversity metric from [6]
>
> We can see in Figure 5 and 16 that our method reliably achieves much higher semantic diversity compared with naive i.i.d. sampling, whose diversity tends to decrease during the online interaction. This suggests that structured brainstorming explores a broader semantic/strategic space rather than merely varying numerical values such as prices. We again thank the reviewer for this constructive suggestion, which significantly strengthens our quantitative analysis.
>
> > Q4: The number of candidates N is set to 5 by default. However, the paper lacks investigation of how hyperparameters, e.g., N, impact the performance and the computational cost. The analysis of hyperparameter sensitivity should be considered for practical deployment, as users need to know how to tune N for their resource constraints.
>
> We thank the reviewer for raising this important issue. As in our response to Q1, we now additionally analyze the sensitivity to the number of candidates $N$ along two dimensions. First, as discussed in our response to Q1, Table 2 reports the trade-off between performance boost and token usage (in log$_2$ scale) for $N \in \\{2,4,6,8,10\\}$ across four different base models, showing that performance improves as $N$ increases. Second, Figure 15 in the appendix reports learning curves for different choices of $N$. These curves show that increasing $N$ consistently improves performance, with diminishing returns beyond moderate values. Together, these results provide practical guidance for choosing $N$ under resource constraints.
>
> We greatly appreciate Reviewer LZah's valuable feedback and constructive suggestions. We hope our responses have addressed all the questions and concerns the reviewer had. We are happy to answer any further questions.
>
> Authors
>
>
> ---
>
> References:
>
> [1] Muennighoff, Niklas, et al. "s1: Simple test-time scaling." Proceedings of the 2025 Conference on Empirical Methods in Natural Language Processing. 2025.
>
> [2] Brown, Bradley, et al. "Large language monkeys: Scaling inference compute with repeated sampling." arXiv preprint arXiv:2407.21787 (2024).
>
> [3] Fu, Yao, et al. "Improving language model negotiation with self-play and in-context learning from ai feedback." arXiv preprint arXiv:2305.10142 (2023).
>
> [4] Xu, Yuzhuang, et al. "Exploring large language models for communication games: An empirical study on werewolf." arXiv preprint arXiv:2309.04658 (2023).
>
> [5] XiaoPeng Yu, Wanpeng Zhang, and Zongqing Lu. 2025. LLM-Based Explicit Models of Opponents for Multi-Agent Games. In Proceedings of the 2025 Conference of the Nations of the Americas Chapter of the Association for Computational Linguistics: Human Language Technologies (Volume 1: Long Papers), pages 892–911, Albuquerque, New Mexico. Association for Computational Linguistics.
>
> [6] Shaib, Chantal, et al. "Standardizing the measurement of text diversity: A tool and a comparative analysis of scores." arXiv preprint arXiv:2403.00553 (2024).

---

### Official Review · Reviewer_Vpgx · 2025-11-01

**Soundness:** 2
**Presentation:** 3
**Contribution:** 2
**Rating:** 4
**Confidence:** 4

**Summary:**

The paper proposes BoN-oppo-simulation, an inference-time self-improvement framework for repeated strategic interactions. It operationalizes the fictitious-play idea for LLM agents by (1) building an opponent model via in-context learning of past interactions and (2) using best-of-N (BoN) candidate generation with opponent simulation to evaluate and select actions by simulating future trajectories. The approach is evaluated on two repeated negotiation games (buyer–seller and resource exchange), compared to several baselines (including BoN variants and “thinking”/CoT baselines), and shown to yield consistent self-improvement across episodes and across backbone models. The paper contains theoretical motivation (Proposition on non-existence of a dominant fixed policy), methodological description, experiments with several LLM backbones, and discussion of limitations.

**Strengths:**

1. The mapping of fictitious play (belief formation + best response) to an inference-time recipe for LLMs is intuitive and well motivated: keep a time-averaged opponent model and use simulation to compute approximate Q’s for candidate actions. This gives a conceptually clean approach to test-time self-improvement without parameter updates.
2. The paper combines (i) in-context opponent modeling with an OFU (optimism-in-face-of-uncertainty) design in prompts and (ii) a structured “brainstorming → concrete action” candidate generation to increase candidate diversity; both are sensible and practically relevant. The distinction between BoN-oppo, BoN-eval, BoN-simulation (CoT), and BoN-oppo (iid) is useful for isolating which component yields gains.

**Weaknesses:**

1. The mapping to fictitious play is conceptually appealing, and Proposition 4.1 is useful to argue the need for adaptation. However, there is no formal analysis of when the inference-time BoN + opponent simulation reliably approximates best responses in multi-turn natural language games (e.g., error amplification through simulated rollouts when opponent model is imperfect). A short theoretical or empirical analysis of sensitivity to opponent-model error would substantially strengthen claims.
2. The approach uses substantial inference-time compute (N candidates and multi-step simulation). The paper discusses scalability and explores N and l in plots, but it lacks concrete wall-clock cost or token-cost tradeoff tables (how much extra latency per turn for N=5 vs N=10, or simulated tokens produced per candidate). For deployment, such numbers matter.
3. The paper shows some experiments where opponents also use BoN variants, but more systematic evaluation of adversarial or fast-learning opponents (or ablations where the opponent model intentionally lags) would clarify limits. There is only partial evidence for dynamic opponents.

**Questions:**

1. The authors note the limitation to 2-agent settings; extension to larger multi-agent scenarios remains open. What if the task scales to >2 agents?
2. Would a full version of prompt text an exact scoring/evaluation be available to readers? For an inference-time, prompt-sensitive method, full prompt text and exact scoring/evaluation formats are essential. (The appendix references prompts, but these must be easy to copy and run for others.)

---

> ### Author Response · Authors · 2025-11-25
>
> ## To Reviewer Vpgx
> We thank Reviewer Vpgx for the insightful feedback. We are encouraged that Reviewer Vpgx finds that our approach is **intuitive, well motivated, and conceptually clean**, and many techniques proposed are **sensible and practically relevant**. We address Reviewer Vpgx's concerns and questions below:
>
> > Q1: The mapping to fictitious play is conceptually appealing, and Proposition 4.1 is useful to argue the need for adaptation. However, there is no formal analysis of when the inference-time BoN + opponent simulation reliably approximates best responses in multi-turn natural language games (e.g., error amplification through simulated rollouts when opponent model is imperfect). A short theoretical or empirical analysis of sensitivity to opponent-model error would substantially strengthen claims.
>
> - **Theoretical analysis:** We adopt the reviewer's suggestion and add a theoretical claim on how the opponent-model error accumulates and propagates to the value function (expected rewards). In short, if the opponent-model error (measured by total variation distance) can be upper-bounded by $\epsilon_h$ for each opponent step $h \in [H]$ with $P(h)=2$, then the value approximation incurs an error of order $\mathcal{O}\left(\sum_{h\in[H]: P(h)=2} \epsilon_h\right)$. This further implies that the best response computed against the opponent model is $\mathcal{O}(H)\cdot \max_{h\in[H]: P(h)=2} \epsilon_h$-approximately optimal against the ground-truth opponent. The theoretical result thus supports our design choice of leveraging an opponent model: as long as the opponent model achieves relatively low errors, it can provide reasonably accurate value estimates for candidate actions and the corresponding policy optimized against the opponent model will also be approximately optimal against the ground truth opponent. We refer the reviewer to Theorem 4.4 of our revised paper for the formal statement and proof.
>
> - **Empirical analysis:** To demonstrate empirically that a more accurate opponent model yields a stronger policy, we additionally report how accurately the opponent model selects the best candidate (evaluated under the ground truth opponent) during online interaction. Specifically, Figures 11 and 12 show the probability that the candidate ranked highest by the opponent model matches the one that achieves the highest reward against the true opponent. We observe that this accuracy is relatively low at the beginning but steadily increases over episodes, which is consistent with the trends in Figure 3 where the agent’s actual performance is also low initially and improves over time. Together with the theoretical analysis, this provides both formal and empirical evidence that our procedure becomes a better approximate best response as the opponent model improves.

---

> ### Author Response · Authors · 2025-11-25
>
> > Q2: The approach uses substantial inference-time compute (N candidates and multi-step simulation). The paper discusses scalability and explores N and l in plots, but it lacks concrete wall-clock cost or token-cost tradeoff tables (how much extra latency per turn for N=5 vs N=10, or simulated tokens produced per candidate). For deployment, such numbers matter.
>
> We thank the reviewer for highlighting this important point regarding deployment.
>
> - ***Discussions on the computation costs:*** Theoretically speaking, compared with the naive baseline without any inference-time boosts, the ratio of computation costs between our BoN-based methods and baseline methods scales **linearly** as
> $$
> \frac{\text{BoN Costs}}{\text{Baseline Costs}} = \mathcal{O}(H \times N),
> $$
> where the dependency on $H$ (the number of time steps in one episode) arises because BoN-based methods simulate each candidate over a (possibly complete) episode, and the dependency on $N$ arises because we need to evaluate $N$ candidates. **More importantly, for wall-clock latency, the dependency on $N$ can be largely mitigated in practice, since one major benefit of BoN-based methods is that the $N$ candidates can be simulated in parallel given sufficient compute.**
>
> - ***New experimental results on tokens vs. performance trade-off:*** In the revised paper, we have added Table 2, which reports, for four different base models, the average performance boost over 20 repeated runs together with the (log$_2$) number of generated tokens for the baseline and our methods with different $N$. Using the log scale for token counts follows standard practice in inference-time scaling works (e.g., [1, 2]). We see in Table 2 that increasing inference-time computation generally leads to higher performance, and that even relatively small $N$ (e.g., $N=4$ or $6$) already achieves substantial gains across all base models, giving practitioners a tunable budget–performance trade-off.
>
> - **Remarks on the computation costs of typical inference-time framework in the literature:** Finally, we would like to remind the reviewer that existing inference-time scaling methods often use inference-time computation that are orders of magnitude higher than the baseline, e.g., [1, 2]. **Therefore, we view the higher computation cost as an inherent and widely common cost of inference-time scaling approaches, rather than a limitation specific to our method.**
>
> > Q3: The paper shows some experiments where opponents also use BoN variants, but more systematic evaluation of adversarial or fast-learning opponents (or ablations where the opponent model intentionally lags) would clarify limits. There is only partial evidence for dynamic opponents.
>
> We agree that understanding the limits against highly adaptive opponents is important.
>
> Therefore, we would like to first clarify that the opponents considered in our experiments are never stationary:
> - (1) The *default opponent LLMs* maintain a memory of all interactions from previous episodes and are always conditioned on this history. As the history accumulates, their behavior naturally changes over episodes (as noted in our footnote 3).
> - (2) We additionally consider opponents with dynamic budgets/costs, which are re-sampled at the beginning of each episode.
> - (3) We further evaluate against opponents that also use our inference-time techniques.
>
> **Additionally, in response to the reviewer’s suggestions, in the revised paper, we introduced three new families of strongly adaptive opponents, each powered by a distinct learning mechanism:**
>
> - **Opponents using the approach from [3] (AI feedback):** This method introduces an additional critic at the beginning of each episode. The critic maintains all history and provides three high-level suggestions/feedbacks on how to improve rewards in the next episode.
>
> - **Opponents using the approach from [4] (selective experience reflection):** The core idea of [4] is to retrieve one negative experience and several good experiences from the history, and then feed these retrieved examples plus a short suggestion to the acting agent at each decision-making step. We therefore refer to this approach as *selective experience reflection*.
>
> - **Opponents using the approach from [5] (private information prediction):** This method introduces an opponent model that predicts the opponent’s private information, which is then fed into the acting agent to support better decision-making.
>
> We report the results in Figure 6, where we symmetrize the buyer-seller game by letting all methods play both roles (buyer/seller) and start both first and second, and then normalize the rewards so that a value $> 0.5$ indicates outperforming the opponent. Under this evaluation, our method consistently outperforms these fast-learning opponents. We refer the reviewer to Section 5 for the revised discussion of dynamic opponents, and to Section E for more detailed descriptions of these three additional approaches.

---

> > ### Author Response · Authors · 2025-11-25
> >
> > > Q4: The authors note the limitation to 2-agent settings; extension to larger multi-agent scenarios remains open. What if the task scales to >2 agents?
> >
> > **Conceptually, our framework does not rely on there being exactly two agents.** This is since fictitious play naturally extends to multi-player games as documented in the classical game-theory literature [6, 7], by treating *the rest of the population as a joint opponent*. The same idea applies here: in an $M$-agent game, the BoN-oppo-simulation agent would use an opponent model that predicts the joint behavior of the $M-1$ other agents given the interaction history. The BoN + simulation machinery itself is unchanged.
> >
> > The reason we believe scaling to many-agent scenarios requires more effort is primarily on the engineering/implementation side. There are several non-trivial research questions about how to model other agents ***when there are many***. For example, there are multiple options:
> > - (1) **Joint modeling.** Model the $M-1$ other agents as a single joint probability distribution. This is the most expressive representation but suffers from an exponential dependency on the number of agents, a.k.a., *the curse of multi-agency*.
> > - (2) **Individual modeling.** Model the other agents individually. This is highly scalable, as the complexity of the opponent model scales linearly with the number of agents, but it ignores potential correlations among opponents.
> > - (3) **Structured modeling.** Interpolate between the above via a structured representation (e.g., an interaction graph) that captures key dependencies without modeling the full joint distribution.
> >
> > Therefore, we do not intend to overclaim that we have fully solved the setting with many agents, but rather to highlight these exciting modeling questions as promising directions for future work.
> >
> > **Finally and most importantly, the focus of the paper is a principled framework for scaling inference-time computation, whereas the opponent model is a flexible component.** We intentionally restrict ourselves to two-agent negotiations in this work, since two-player settings are already the standard testbed for strategic reasoning in LLMs [8] and allow us to cleanly evaluate inference-time scaling ideas. We therefore believe that extending opponent modeling to larger multi-agent systems is a rich but less relevant direction for future work, rather than a fundamental limitation of our approach.
> >
> > > Q5: Would a full version of prompt text an exact scoring/evaluation be available to readers? For an inference-time, prompt-sensitive method, full prompt text and exact scoring/evaluation formats are essential. (The appendix references prompts, but these must be easy to copy and run for others.)
> >
> > We thank the reviewer for their interest in the implementation details of our framework. For all prompts, we have additionally included them as a Python filesin the anonymous GitHub repository https://github.com/llmnegotiationiclr-anonymous/llm-negotiation, so that they can be easily copied and run. For infrastructure such as the evaluation/scoring format, we simply reuse the gaming environment from NegotiationArena [8], as described in the main text.
> >
> > We greatly appreciate Reviewer Vpgx's valuable feedback and constructive suggestions. We hope our responses have addressed all the questions and concerns raised. We are happy to answer any further questions.
> >
> > Authors
> >
> >
> > ---
> >
> > References:
> >
> > [1] Muennighoff, Niklas, et al. "s1: Simple test-time scaling." Proceedings of the 2025 Conference on Empirical Methods in Natural Language Processing. 2025.
> >
> > [2] Brown, Bradley, et al. "Large language monkeys: Scaling inference compute with repeated sampling." arXiv preprint arXiv:2407.21787 (2024).
> >
> > [3] Fu, Yao, et al. "Improving language model negotiation with self-play and in-context learning from ai feedback." arXiv preprint arXiv:2305.10142 (2023).
> >
> > [4] Xu, Yuzhuang, et al. "Exploring large language models for communication games: An empirical study on werewolf." arXiv preprint arXiv:2309.04658 (2023).
> >
> > [5] Yu, XiaoPeng, Wanpeng Zhang, and Zongqing Lu. "LLM-Based Explicit Models of Opponents for Multi-Agent Games." Proceedings of the 2025 Conference of the Nations of the Americas Chapter of the Association for Computational Linguistics: Human Language Technologies (Volume 1: Long Papers). 2025.
> >
> > [6] Monderer, Dov, and Lloyd S. Shapley. "Fictitious play property for games with identical interests." Journal of economic theory 68.1 (1996): 258-265.
> >
> > [7] Sela, Aner. "Fictitious play in ‘one-against-all’ multi-player games." Economic Theory 14.3 (1999): 635-651.
> >
> > [8] Bianchi, Federico, et al. "How Well Can LLMs Negotiate? NegotiationArena Platform and Analysis." International Conference on Machine Learning. PMLR, 2024.

---

### Author Response · Authors · 2025-11-25
**Summary of our revision and responses**

# Summary of our rebuttal efforts and important clarifications

We first thank all reviewers for their detailed and constructive feedback. We are encouraged that reviewers share similar positive assessments that our framework is **clean, principled, and conceptually appealing**, that our theoretical justifications are **substantial**, and that our implementations are **granular**. Below we summarize and highlight our main revision efforts during the rebuttal (with new contents in the manuscript colored in blue), aimed at addressing all reviewers’ concerns.

### Paper revision: new experiments

- **New experiments on computation analysis in Table 2.**
  We now provide a quantitative analysis of inference-time computation, reporting for four different base models the average performance boost over 20 runs together with the log$_2$ number of generated tokens for varying candidate counts $N$. This explicitly characterizes the performance–compute trade-off of our BoN-based methods.

- **New experiments on testing against more strongly adaptive opponents powered by three new baselines in Figure 6.**
  We additionally evaluate our approach against three families of fast-learning opponents instantiated from prior work (AI feedback, selective experience reflection, and private-information prediction). The new results demonstrate that our BoN–opponent-simulation agent continues to self-improve and reliably outperforms these adaptive baselines.

- **New experiments on evaluating semantic diversity in Figures 5 & 16.**
  Beyond numeric dispersion, we now report semantic diversity metrics for candidate messages, including embedding-based diversity and 1–Self-BLEU. These experiments show that our structured brainstorming procedure generates consistently more diverse candidates than i.i.d. sampling throughout online interaction.

- **New experiments on opponent architecture generalization in Table 3.**
  We further study cross-architecture robustness by letting our Gemini-based agent negotiate against opponents powered by Claude, Qwen, and LLaMA. The new table shows that our method continues to achieve the largest performance gains over baselines even when the true opponent and the opponent model use different LLM architectures.

### Paper revision: new theoretical analysis

- **We now provide Theorem 4.4.**
  The new theorem formally analyzes how errors in the opponent model, measured in total variation distance at each opponent step, propagate to value estimates and the resulting policy. It shows that the value approximation error and the approximate optimality gap scale linearly with the accumulated per-step modeling errors, thereby theoretically justifying the use of an opponent model for BoN-based candidate evaluation.

### Paper revision: new discussions and references

- **More proper references and discussions.**
  We have included some additional related works as required by Reviewer oQJh. We also explicitly position our framework relative to these lines of work.

- **More discussions on experimental environments.**
  We clarify why our two negotiation games are an appropriate and informative testbed for strategic reasoning with LLMs, emphasizing their mixed-motive nature, private information, multi-turn natural-language action space, and relevance as standard benchmarks in the literature.

### Important clarifications

- **Clarifications on the problem focus of our paper.**
  We clarify that our primary focus is *no-regret–style test-time self-improvement/adaptation* in repeated interactions with unknown opponents, rather than equilibrium or non-exploitable policy seeking during training. Our fictitious-play–inspired analysis connects our framework to online learning and highlights that the goal is to obtain an adaptive sequence of policy that improves performance across episodes using only inference-time computation, not to train a single non-exploitable policy.

- **Clarifications on our (output-level) inference-time scaling vs. (input-level) prompt/context engineering vs. fine-tuning.**
  We further distinguish three complementary axes: (i) **output-level inference-time scaling** (our BoN + multi-step opponent simulation, operating on the model’s output distribution without parameter updates), (ii) **input-level prompt/context engineering** (including AI feedback and reflection-style methods that modify prompts and contexts while decoding a single trajectory), and (iii) **fine-tuning / self-play training** (which updates model parameters offline). We emphasize that our contributions lie in the output-level inference-time regime, and we clarify that these approaches are orthogonal and potentially combinable rather than competing.

Again we thank all the reviewers for their constructive feedbacks and acknowledge that it has helped us revised our paper significantly!

---

### Meta-Review · Area_Chair_TGdU · 2026-01-06

**Summary:**

This work introduce an inference-time framework called best-of-sampling with opponent simulation. Basically, the idea lacks of novelty. In literature of self-play (self-improvement) of game playing, there are many solutions on that, such as:
Strategist: Self-improvement of LLM Decision Making via Bi-Level Tree Search. ICLR'25
"Limitted novelty" is also raised by reviewer oQJh.

In the rebuttal, 3 reviewers are negative, 1 is positive. From the rebuttal, no reviewers have been convinced to update the score.

**Reviewer Concerns:**

1. Vpgx, oQJh
Framework is evaluated only in 2-agent settings; scalability to many agents is unclear.
The rebuttal basically does not add additional exp to validate it. Zero experiments. No concrete instantiation.

2. "Limitted novelty" is also raised by reviewer oQJh.
This critical concern is not well-addressed in the rebuttal.

3. Missing or inadequate external baselines. It is raised by oQJh (strongly), partially by others.
Concerns follow-up that results are “insular,” tested only against in-house or weak baselines. Some baselines are clearly needed, such as:
Strategist: Self-improvement of LLM Decision Making via Bi-Level Tree Search. ICLR'25

**Reviewer Scores:**

In the rebuttal, 3 reviewers are negative, 1 is positive. From the rebuttal, no reviewers have been convinced to update the score.

---

### Decision · Program_Chairs · 2026-01-26

Reject